eLife

# Systematic identification of *cis*-regulatory variants that cause gene expression differences in a yeast cross

**Kaushik Renganaath[1†], Rockie Chong[2†], Laura Day[3,4,5], Sriram Kosuri[2], Leonid Kruglyak[3,4,5]\*, Frank W Albert[1]\***

[1]Department of Genetics, Cell Biology, & Development, University of Minnesota, Minneapolis, United States; [2]Department of Chemistry & Biochemistry, University of California, Los Angeles, Los Angeles, United States; [3]Department of Human Genetics, University of California, Los Angeles, Los Angeles, United States; [4]Department of Biological Chemistry, University of California, Los Angeles, Los Angeles, United States; [5]Howard Hughes Medical Institute, University of California, Los Angeles, Los Angeles, United States

**Abstract** Sequence variation in regulatory DNA alters gene expression and shapes genetically complex traits. However, the identification of individual, causal regulatory variants is challenging. Here, we used a massively parallel reporter assay to measure the *cis*-regulatory consequences of 5832 natural DNA variants in the promoters of 2503 genes in the yeast *Saccharomyces cerevisiae*. We identified 451 causal variants, which underlie genetic loci known to affect gene expression. Several promoters harbored multiple causal variants. In five promoters, pairs of variants showed non-additive, epistatic interactions. Causal variants were enriched at conserved nucleotides, tended to have low derived allele frequency, and were depleted from promoters of essential genes, which is consistent with the action of negative selection. Causal variants were also enriched for alterations in transcription factor binding sites. Models integrating these features provided modest, but statistically significant, ability to predict causal variants. This work revealed a complex molecular basis for *cis*-acting regulatory variation.

**\*For correspondence:**
LKruglyak@mednet.ucla.edu (LK);
falbert@umn.edu (FWA)

[†]These authors contributed equally to this work

**Competing interests:** The authors declare that no competing interests exist.

## Introduction

Individual genomes carry thousands of sequence differences in gene-regulatory elements. Collectively, these variants contribute to variation in many phenotypic traits by altering the expression of one or multiple genes (*Albert and Kruglyak, 2015*). The presence of individual DNA variants that alter gene expression can be detected by mapping genomic regions called 'expression quantitative trait loci' (eQTLs). Among these, 'local' eQTLs are located close to or in the gene whose expression they influence. Eukaryotic species ranging from yeast to human carry large amounts of local regulatory variation (*Brem et al., 2002*; *Hasin-Brumshtein et al., 2016*; *Heyne et al., 2014*; *Rockman et al., 2010*; *Stranger et al., 2005*; *West et al., 2007*). In human populations, most genes are affected by one or multiple local eQTLs (*GTEx Consortium et al., 2017*). Similarly, in a cross between two genetically different yeast isolates, 74% of genes are influenced by local eQTLs (*Albert et al., 2018*). Most of these local eQTLs arise from DNA variants that perturb *cis*-acting regulatory mechanisms (*Albert et al., 2018*; *Ronald et al., 2005*). When such *cis*-acting variants are located in a gene's transcribed region, they may alter mRNA stability, splicing, polyadenylation, or regulation by RNA-binding proteins. *Cis*-acting variants in promoters or enhancers may alter the transcription of their target genes.

As a consequence of genetic linkage in experimental crosses (*Albert et al., 2018*) and linkage disequilibrium in outbred populations (*GTEx Consortium et al., 2017*; *Kita et al., 2017*), regions mapped as eQTLs almost always contain multiple sequence variants. Typically, it is assumed that most of these variants have no effect, obscuring the identity of one or a few causal variants in each eQTL (*Figure 1*). Although the causal variants in several local eQTLs have been identified (*Chang et al., 2013*; *Claussnitzer et al., 2015*; *Lutz et al., 2019*; *Musunuru et al., 2010*; *Ronald et al., 2005*), most causal variants remain unknown. Because of this lack of systematic information, many questions about local eQTLs remain open, including whether local eQTLs are typically caused by one or multiple variants, if multiple variants interact in a non-additive fashion, what evolutionary forces act on causal variants, which molecular mechanisms causal variants perturb, and whether it may ultimately be possible to combine the answers to these questions to construct models that can predict the consequences of regulatory variants from genome sequence.

Massively parallel reporter assays (MPRAs) have begun to make it possible to dissect regulatory activity of DNA sequences at scale (*Kinney et al., 2010*; *Melnikov et al., 2012*; *Mulvey et al., 2020*; *Patwardhan et al., 2009*). In these approaches, which have been applied in bacteria (*Cambray et al., 2018*; *Kinney et al., 2010*; *Kosuri et al., 2013*), yeast (*Cuperus et al., 2017*; *Mogno et al., 2013*; *Sharon et al., 2012*), flies (*Gisselbrecht et al., 2013*), zebrafish (*Rabani et al.,*

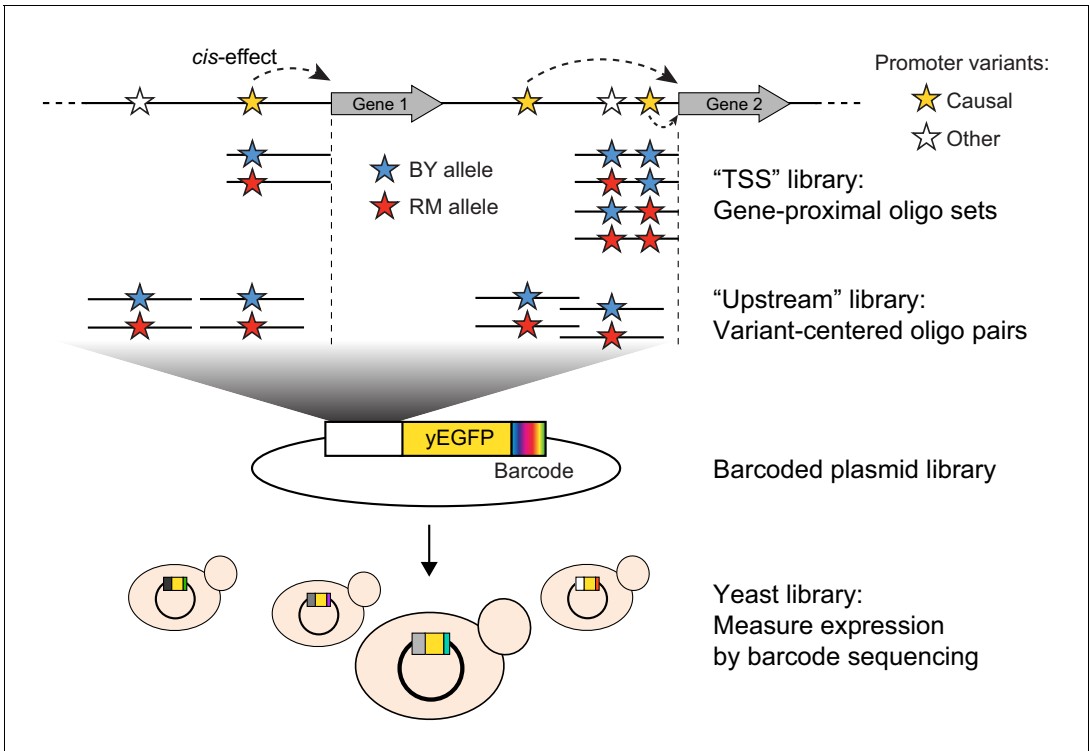

**Figure 1.** Schematic of MPRA design. At the top, two native genes in the genome with multiple promoter variants (stars) are shown. Below the two genes, the TSS and Upstream MPRA library designs are illustrated. The TSS library tests all variants within 144 bp of the transcription start site (dashed vertical lines), while the Upstream library tests a subset of TSS variants along with variants located further away from the transcription start site.

The online version of this article includes the following figure supplement(s) for figure 1:

**Figure supplement 1.** Size distribution of indels in the MPRA design.

**Figure supplement 2.** Distribution of the number of promoter variants per gene in the MPRA design.

**Figure supplement 3.** Schematic of the library cloning procedure.

**Figure supplement 4.** Distributions of barcodes.

**Figure supplement 5.** Number of times a given designed oligo was observed in the TSS annotation sequencing run as a function of the first two nucleotides of the oligo.

**Figure supplement 6.** Barcode amplification.

**Figure supplement 7.** Reproducibility of oligo expression.

**Figure supplement 8.** Correlations of expression driven by 200 oligos common to the TSS and Upstream library.

*2017*), mouse tissues (*Kwasnieski et al., 2014*; *Smith et al., 2013*), and cultured human cells (*Mulvey et al., 2020*), pooled libraries of DNA oligos are placed next to or in a reporter gene and inserted into populations of cells. The activity of each oligo is then assayed in bulk by pooled high-throughput sequencing. MPRAs have been performed with synthesized libraries of designed oligos (*Sharon et al., 2012*), with fragmented genomic DNA (*van Arensbergen et al., 2019*; *Arnold et al., 2013*; *Wang et al., 2018*), and with randomly generated DNA (*Cuperus et al., 2017*; *de Boer et al., 2020*; *Rosenberg et al., 2015*). MPRA readouts have included sequencing of either the oligos themselves (*Arnold et al., 2013*) or short barcodes that tag each oligo (*Kwasnieski et al., 2012*). MPRAs have quantified mRNA expression by sequencing cDNA (*Kwasnieski et al., 2012*) and protein expression based on 'FACS-Seq' approaches in which cells are sorted into bins of increasing fluorescent reporter gene activity (*Kinney et al., 2010*; *Matreyek et al., 2018*; *Sharon et al., 2012*). MPRAs have been conducted using plasmid-borne reporters as well as reporters integrated into the genome (*Inoue et al., 2017*; *Maricque et al., 2019*; *Mogno et al., 2013*).

MPRAs have been used to probe DNA sequences for their ability to drive transcription (*Arnold et al., 2013*; *Kheradpour et al., 2013*; *Wang et al., 2018*), dissect the importance of individual bases in regulatory elements (*Patwardhan et al., 2009*), and examine the combined effects of multiple elements in regulatory 'grammars' (*Davis et al., 2020*; *Kosuri et al., 2013*; *Mogno et al., 2013*; *Sharon et al., 2012*; *Smith et al., 2013*) in promoters (*Kotopka and Smolke, 2020*; *Lubliner et al., 2015*; *Sharon et al., 2012*; *Weingarten-Gabbay et al., 2019*), UTRs (*Cuperus et al., 2017*; *Dvir et al., 2013*; *Rabani et al., 2017*; *Shalem et al., 2015*) and enhancers (*Arnold et al., 2013*; *Klein et al., 2019*; *Melnikov et al., 2012*; *Patwardhan et al., 2009*). Other applications have assayed sequences that promote splicing (*Cheung et al., 2019*; *Rosenberg et al., 2015*), translation (*Goodman et al., 2013*; *Weingarten-Gabbay et al., 2016*), DNA methylation (*Krebs et al., 2014*) and RNA editing (*Safra et al., 2017*). More recently, MPRAs have identified individual human DNA variants that alter gene expression (*Tewhey et al., 2016*; *Ulirsch et al., 2016*) in studies ranging in scale from variants in specific regions implicated by genome-wide association studies for a given disease (*Choi et al., 2019*; *Liu et al., 2017*; *Pashos et al., 2017*; *Vockley et al., 2015*) to a genome-wide survey of nearly six million common single-nucleotide polymorphisms (*van Arensbergen et al., 2019*). In spite of these successes, the size of the human genome, which harbors tens of millions of rare as well as common variants (*1000 Genomes Project Consortium et al., 2015*), combined with a high degree of tissue-specificity in gene expression and the activity of regulatory DNA *GTEx Consortium et al., 2017*; *Inoue et al., 2019*; *Maricque et al., 2019* have complicated dissection of causal variants in local eQTLs. The compact gene-regulatory regions of *S. cerevisiae*, combined with comprehensive eQTL maps (*Albert et al., 2018*), provide an excellent opportunity to study regulatory variants systematically.

Here, we used an MPRA to probe thousands of intergenic variants that differ between two yeast isolates. We identified 451 variants with significant *cis*-acting effects on mRNA expression. These causal variants underlie known local eQTLs. We found that individual local eQTLs can harbor multiple causal variants, including pairs of variants with non-additive, epistatic effects. Causal variants tended to alter transcription factor binding motifs and showed signs of evolving under negative selection. Combinations of these features predicted causal variants better than expected by chance, albeit with modest accuracy.

## Results

### An MPRA to assay *cis*-regulatory variants in yeast promoters

At least half of the genes in the yeast genome are influenced by local eQTLs that segregate between the laboratory strain BY, a close relative of the genome reference strain S288C, and RM, a vineyard isolate that is related to strains commonly used in wine making (*Peter et al., 2018*). Most of these eQTLs act in *cis* (*Albert et al., 2018*), making the BY and RM strains a rich reservoir for identifying causal *cis*-acting variants. Here, we studied DNA variants in yeast promoters, which we defined as the intergenic region upstream of the transcription start site of a given gene up to the coding region of the adjacent gene, for a maximum of 1000 bases. These promoter regions differ between BY and RM at 11,768 single-nucleotide variants (SNVs) and 2442 insertion/deletion variants (indels) in 3176 genes (*Bloom et al., 2013*).

To assay the effects of individual variants on gene expression, we designed an MPRA composed of two synthetic promoter libraries encoded by pooled oligonucleotides (*Figure 1*, *Table 1*, *Supplementary file 1*). The 'TSS' library assayed all variants in the 144 nucleotides immediately upstream of the transcription start site (*Pelechano et al., 2013*), a region that is highly enriched for transcription factor binding sites (*Lin et al., 2010*). When multiple variants were present within this region for a given gene, we designed one sequence that carried the BY allele at all variants, one sequence that carried the RM allele at all variants, and a set of sequences that each carried the RM allele at a single variant and the BY allele at all other variants. The 'Upstream' library mostly assayed variants located further upstream of the transcription start site than those in the TSS library, using a pair of sequences that represented 144 bp of genomic DNA centered on the given variant. By design, the TSS and Upstream libraries shared a subset of variants. Together, they assayed 7005 unique variants (5,758 SNVs and 1247 indels; *Figure 1—figure supplement 1*) in the promoters of 3076 genes, with a median of two variants per gene (*Figure 1—figure supplement 2*).

The synthesized oligo libraries were placed upstream of a yEGFP reporter gene (*Sheff and Thorn, 2004*) on a low-copy number yeast plasmid (*Figure 1*). Prior to adding the yEGFP gene, we added random barcodes with a length of 20 nucleotides downstream of the reporter gene (*Figure 1* and *Figure 1—figure supplement 3*). These barcodes were expressed as part of the 3' end of the reporter mRNA, such that barcode abundance provided a measure of gene expression driven by the given oligo. We used paired-end sequencing to map each barcode to the oligo it tagged (*Figure 1—figure supplement 3*). Most oligos were tagged by hundreds of barcodes (*Table 1*, *Figure 1—figure supplement 4*) to control for possible influences of the expressed barcodes on mRNA levels. The final plasmid libraries we used in our experiments contained more than 90% of the designed oligos, which assayed 5832 variants in the promoters of 2503 genes (*Table 1*; see Materials and methods and *Figure 1—figure supplement 5* for a description of oligos lost during oligo synthesis and/or cloning). The libraries were transformed into the BY strain and grown in independent replicate cultures (*Table 1*). We grew each culture to late exponential phase, extracted mRNA and plasmid DNA from each culture, and sequenced barcodes (*Figure 1—figure supplement 6*) to a median depth of 46 million RNA reads (range 19–70 million) and 31 million DNA reads (range 14–84 million) per sample (*Table 1—source data 1*).

## Oligo expression is reproducible and reflects gene expression in the genome

We conducted three analyses to assess the reliability of our data. First, we measured the reproducibility of oligo-driven reporter expression. We summed the RNA counts of all barcodes assigned to a given oligo (*Supplementary file 2*), divided them by the respective summed DNA counts to normalize for unequal library composition, and $\log_2$-transformed the resulting ratio. Spearman's rank correlation coefficients (rho) among pairs of replicates ranged from 0.79 to 0.90 (median = 0.83; all

**Table 1.** Library design and representation in experiments.

| Library | TSS | Upstream |
|---|---|---|
| Designed oligos | 7211 | 9882 |
| Variants in design | 3645 | 4547 |
| Genes in design | 2172 | 1918 |
| Oligos in finished library | 6565 | 9646 |
| Barcodes | 9.2 million | 20 million |
| Median barcodes per oligo | 590 | 1008 |
| Variants with data | 2427 | 4467 |
| Genes with data | 1429 | 1824 |
| Number of replicates | 12 | 6 |

The online version of this article includes the following source data for Table 1:
**Source data 1.** Information on replicate samples.

p<2.2e-16) in the Upstream library and from 0.55 to 0.93 (median = 0.74; all p<2.2e-16) in the TSS library (*Figure 1—figure supplement 7A & D*).

Second, our two libraries included a common set of 200 oligos whose sequences we had sampled from a prior study (*Sharon et al., 2012*), allowing us to determine reproducibility between replicates, libraries, and with prior work. Median correlations between TSS and Upstream replicates (rho = 0.76; all p<2.2e-16) were nearly as high as those between replicates within each library (TSS: 0.8, Upstream: 0.99; all p<2.2e-16; *Figure 1—figure supplement 8*). Further, the 200 oligos showed significant, positive correlations with their published expression levels (rho $\geq$ 0.4, p$\leq$1e-6), in spite of experimental and design differences between studies (*Figure 1—figure supplement 7B & E*).

Third, we asked if the promoter fragments in our plasmid libraries were able to recapitulate the expression of genes in their native genomic locations. To do so, we computed the average expression driven by all oligos extracted from the promoter region of a given gene. Although each oligo contained at most 150 bp of promoter sequence on a plasmid, expression in both libraries correlated significantly (rho $\geq$ 0.23, p<2.2e-16) with the native mRNA levels of genes in the genome (*Figure 1—figure supplement 7C & F*). In sum, our assay quantified the regulatory activity of promoter fragments in a manner that was reproducible within our study and when compared to earlier work, and that reflected the activity of native promoters in the genome.

## Identification of hundreds of *cis*-acting promoter variants

The median fold change between alleles was 1.4-fold, and only 29 causal variants (6%) altered expression by more than two-fold (*Figure 2A*). These magnitudes are in good agreement with known genetic effects in the BY/RM strains, in which the vast majority of local eQTLs (*Albert et al., 2018*) and allele-specific, *cis*-acting effects on expression (*Albert et al., 2014a*) alter mRNA abundance by less than 2-fold.

Our MPRA assayed some variants multiple times in different sequence contexts, and we used this redundancy to assess the reproducibility of variant effects. First, 359 variants that reside in the intergenic region between two divergently expressed genes were tested twice in the same library, but in opposite orientation relative to the reporter gene (*Figure 2—figure supplement 1A*). Variants that were significant in at least one strand orientation were likely to also be significant in the other orientation (Fisher's exact test (FET): p=0.0003, odds ratio (OR) = 7). These significant variants also agreed well in the direction of variant effect, that is whether the RM allele drives higher or lower expression than the BY allele (FET: p=0.007, OR = 6.1). Second, 527 variants were assayed in both the TSS and the Upstream libraries, where they were embedded in oligos that had the same strand orientation but differed in the exact window of DNA surrounding each variant (*Figure 2—figure supplement 1B*). Among these, variants that were significant in at least one library were likely to also be significant in the other (FET p=0.007, OR = 4), with significant directional agreement (FET p=0.046, OR = 2.9). Thus, in spite of the small effects of natural sequence variants, our assay was able to reproducibly identify individual causal variants.

Our assay identified several known causal variants. For example, a variant in the promoter of the *OLE1* gene affects *OLE1* expression in *cis* and is likely to be the single causal variant in this promoter (*Lutz et al., 2019*). This variant was highly significant in the MPRA (*Figure 2B*), while a second variant in the *OLE1* promoter was not (*Figure 2A*).

In the promoter of the *SFA1* gene, the MPRA detected a single causal variant out of nine that were assayed (*Figure 2A*). The RM allele of the causal variant is also present in a strain used for bioethanol production (YJS329), where it has been proposed to increase *SFA1* expression by creating an Msn2/4 binding site (*Maurer et al., 2017*; *Zheng et al., 2012*). In our assay, this allele increased gene expression significantly (*Figure 2C*). Taken together, these results show that our MPRA reproducibly detected hundreds of individual *cis*-acting DNA variants.

## Local eQTLs can be caused by single or multiple causal variants

The identity of causal variants in QTL regions is a major question in the genetics of complex traits. We asked if the variants identified by our MPRA underlie the many local eQTLs that segregate between the BY and RM strains. Specifically, we turned to local eQTLs mapped in 1012 recombinant individuals obtained by crossing BY and RM (*Albert et al., 2018*). Because of genetic linkage in the cross, the effects of neighboring variants create one aggregated, spread-out signal of local variation

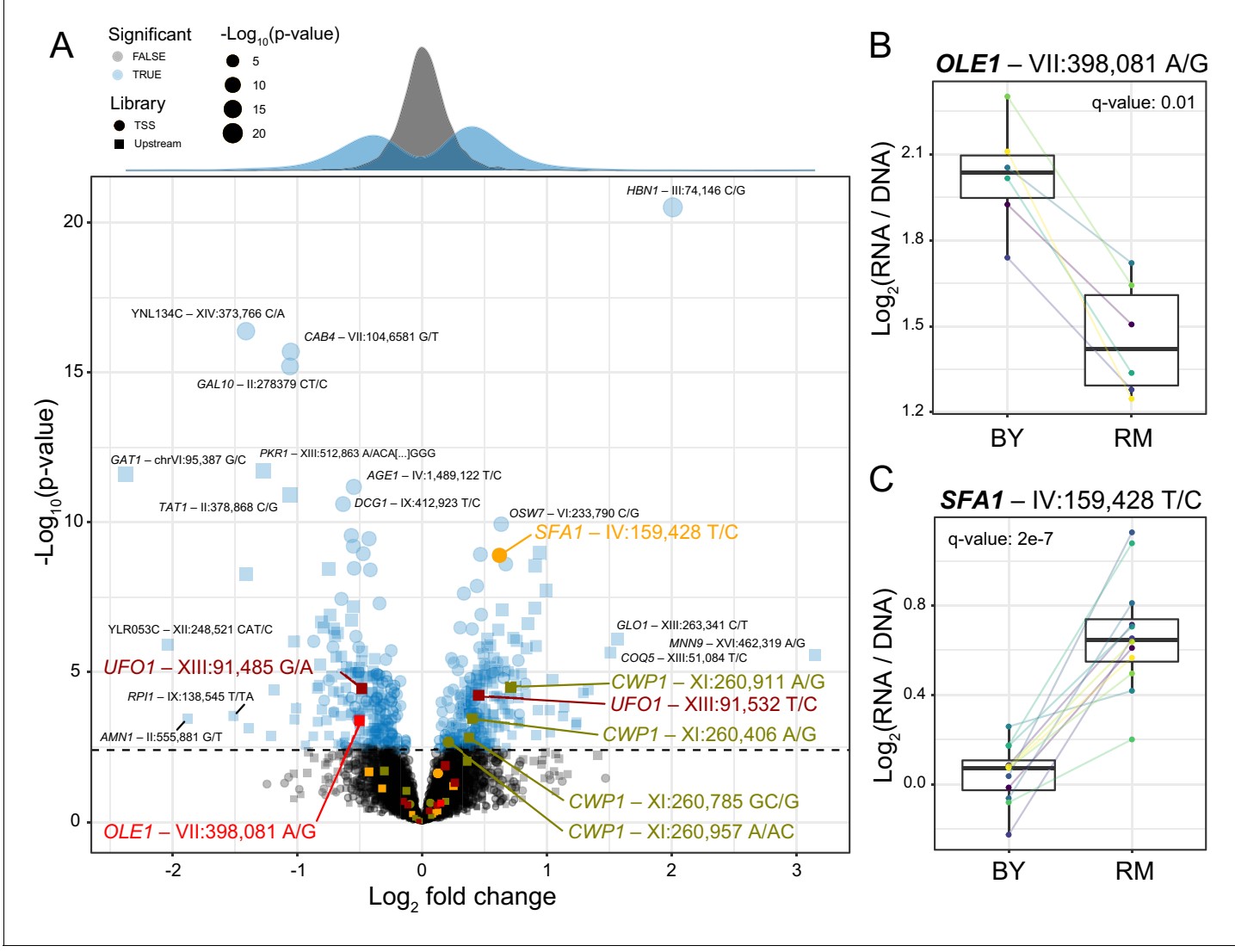

**Figure 2.** Identification of single causal variants. (A) A scatterplot showing the effect size and significance for each variant. The genome-wide significance threshold is shown as a dashed horizontal line. Variants with the most significant effects, along with the genes they affect, are indicated. Variants and genes highlighted in color are described in the text. The histograms at the top shows the distribution of effect sizes for causal (blue) and non-causal (gray) variants. (B) A variant in the promoter of OLE1 known to affect OLE1 expression has a significant effect in the Upstream MPRA. The figure shows expression values for oligos carrying the two alleles. Colored lines and dots indicate different biological replicate experiments. Boxplots show the median as thick line, with the box showing the 25th and 75th percentiles. Whiskers show the largest value no further than 1.5 times the inter-quartile range; points beyond this range are shown as individual points. (C) As in (B) for a variant in the SFA1 promoter, which was detected in the TSS library. To identify individual causal variants, we tested each promoter variant for its effect on reporter gene expression. We detected 166 variants with significant effects in the TSS library and 293 variants in the Upstream library at a false discovery rate (FDR) of 5% (*Figure 2—source data 1*). The $\pi_1$ statistic (*Storey and Tibshirani, 2003*) computed across all variants suggested that at least 26 and 31% of variants had effects on gene expression in the TSS and Upstream libraries, respectively, even if these variants could not all be detected with individual significance. There were 451 unique variants that reached significance across the two partially overlapping libraries (A, *Figure 2—source data 2*).

The online version of this article includes the following source data and figure supplement(s) for figure 2:

**Source data 1.** Statistical tests for each variant.

**Source data 2.** Statistical results for each variant after aggregating across the two sub libraries.

**Figure supplement 1.** Reproducibility of variant effects.

at each gene. For 2884 genes, these effects were strong enough to be detected as local eQTLs at genome-wide significance (*Albert et al., 2018*). We asked to what extent these local eQTLs can be explained by individual causal variants identified here (*Figure 3—source data 1*).

Initially, we considered all 2300 genes with available data from both the cross and the MPRA, irrespective of whether their local effect had reached significance in the cross. Likewise, we summed the MPRA effects of all assayed variants for a given gene, irrespective of their significance. There was a significant correlation between local eQTL effects and summed MPRA effects (rho = 0.1, p=1e-6). This overall correlation is likely degraded by noise in both datasets, in particular for small, non-significant effects. Therefore, we first restricted our analyses to variants with significant (5% FDR) MPRA effects. The summed effects of these significant variants showed a stronger correlation with eQTL effects than did those of all variants (rho = 0.24, p=5e-6). Next, to also avoid noise in the eQTL effect estimates, we further restricted the comparison to the 238 genes with strong local eQTLs— those with a LOD score of at least 50. This filter again improved the correlation with summed significant MPRA effects (rho = 0.47, p=2e-5). Allele-specific expression in diploid hybrids provides an independent estimate of the aggregated effect of all *cis*-acting variants that affect a given gene (*Emerson et al., 2010*; *Ronald et al., 2005*; *Wittkopp et al., 2004*). Genes with significant allele-specific expression in published BY/RM hybrid data (*Albert et al., 2014a*; *Albert et al., 2018*) were more likely to have a causal variant in our MPRA (FET: OR = 3.2, p<2.2e-16). These agreements show that, in aggregate, our MPRA successfully captured effects of variants that cause local, *cis*-acting eQTLs.

We asked whether single MPRA variants could account for local eQTLs. At each gene, we extracted the most significant variant among those with an FDR of ≤5%. The effects of these single variants correlated with strong local eQTLs (rho = 0.41, p=3e-4) and showed significant agreement between the direction of the effects of the variants and the eQTLs (FET: OR = 6.9, p=9e-5; *Figure 3A*). For example, the MPRA effect for the single causal variant in the *OLE1* promoter (*Lutz et al., 2019*) recapitulated the magnitude of the *OLE1* local eQTL almost perfectly (*Figure 3A & C*). Thus, we identified individual causal variants that underlie local eQTLs.

QTL regions for organismal traits can harbor multiple causal variants (*Mackay et al., 2009*), but it is less clear whether QTLs for cellular traits such as gene expression have a similarly complex molecular basis. To test whether local eQTLs contain multiple causal variants, we considered, where available, the second-most significant MPRA variant for each gene, if that variant was still significant at an FDR of 5%. The effects of these second variants correlated with eQTL effects (rho = 0.3, p=0.03; *Figure 3B*). Further, there were significant correlations between the number of nominally significant (p<0.05) MPRA variants identified for a gene and that gene's local eQTL LOD score (rho = 0.12, p=0.003) and its absolute eQTL effect size (rho = 0.14, p=3e-5). Thus, some local eQTLs are due to multiple causal promoter variants.

Some genes were affected by more than two causal variants. For example, we detected four causal variants at the *CWP1* gene (*Figure 2A*). At each of these variants, the RM allele increased expression, and their summed effects approximated that of the *CWP1* local eQTL (*Figure 3C*).

Some genes had two significant variants with opposite direction of effect. For example, we detected two variants located 47 bases apart in the promoter of the *UFO1* gene that had effects of similar magnitude but in opposite direction (*Figure 2A*). *UFO1* does not have a significant local eQTL (*Figure 3C*), suggesting that this gene is influenced by two variants whose effects cancel each other out.

Together, these observations suggest that the molecular basis of local eQTLs spans a range of scenarios. Some local eQTLs are due to single causal variants, while others arise from joint effects of multiple causal variants. When multiple variants have effects in the same direction, they sum to create a stronger eQTL. When their effects are in the opposite directions, they may be invisible in a cross.

## The effects of individual variants may be influenced by nucleosome binding

Our plasmid-based MPRA tested variants in a molecular context that differs from the native genomic context in two respects: in the DNA sequence beyond that present on the oligos and, potentially, in the chromatin state in which each variant is embedded. Specifically, nucleosomes influence gene expression (*Han and Grunstein, 1988*), and may mask or otherwise alter the effects of variants they

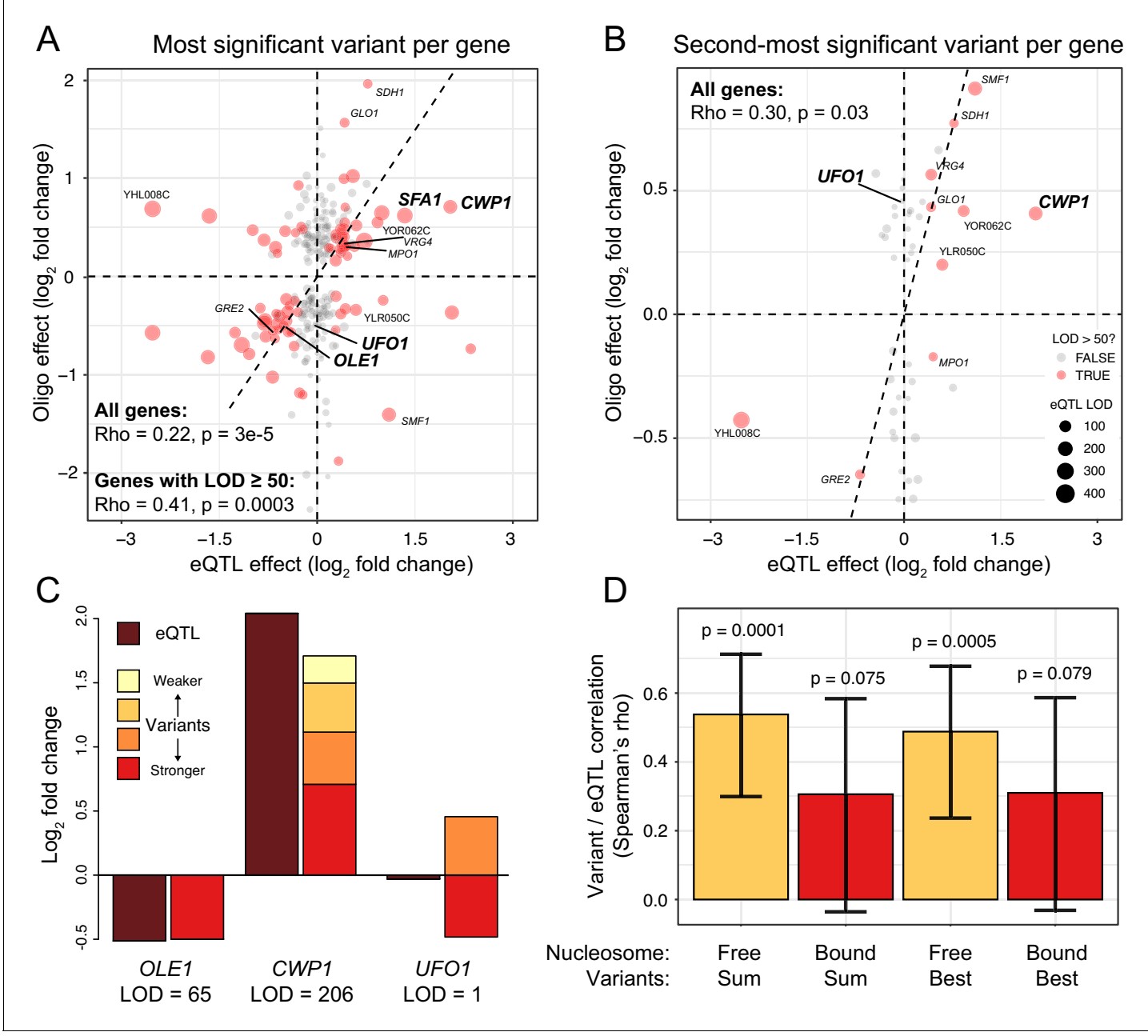

**Figure 3.** Comparison of variant effects to local eQTLs. (**A**) Scatterplot showing the MPRA effect of the most significant causal variant per gene (y-axis) versus the effect of the local eQTL (x-axis). Red dots indicate local eQTLs with a LOD score of at least 50. Genes describes in the text and in panel (**C**) are highlighted in bold. Other genes also present in panel (**B**) are indicated in regular font. The dashed diagonal shows the case of equal MPRA and local eQTL effects. Dashed horizontal and vertical lines indicate no effect. The panel only shows variants with a significant MPRA effect. (**B**) As in (**A**), but for the second-most significant causal variant per gene. Note the different scale of the y-axis between A and B. (**C**). Examples of summed MPRA effects of individual variants compared to the local eQTL for the given gene. (**D**). Spearman correlation coefficients MPRA variants versus local eQTLs as a function of whether a variant is bound by a nucleosome in the genome. Significance of the correlation is indicated. Error bars show 95% confidence intervals for the strength of the correlation.

The online version of this article includes the following source data for figure 3:

**Source data 1.** Local eQTLs and variant results.

bind (*Rando and Winston, 2012*; *Zhu et al., 2018*). We asked whether the effects of variants located in nucleosome-free regions in the genome (*Brogaard et al., 2012*) were better captured by our MPRA than those of nucleosome-bound variants. Indeed, the summed effects of significant variants in nucleosome-free regions correlated better with strong local eQTLs than those of nucleosome-bound variants, which showed no significant correlation (*Figure 3D*). A similar difference was seen for the single most significant MPRA variant (*Figure 3D*). This better agreement between the genome and the MPRA for the effects of variants in nucleosome-free regions would be expected if these variants are also not bound by nucleosomes on the plasmids, and are thus exposed to a molecular environment that resembles that in the genome. More broadly, this result suggests that nucleosome occupancy and chromatin state may alter the regulatory effects of genomic variants, consistent with MPRAs in human cells (*Inoue et al., 2019*; *Maricque et al., 2019*) and with the fact that many human local eQTLs vary among tissues and cell types (*GTEx Consortium et al., 2017*; *Yao et al., 2020*).

## Non-additive interactions among promoter variants

The presence of multiple causal variants in single promoters raises the question of whether these variants have independent, additive effects or if the effects of some variants are modulated by the presence of the other variant(s) in a non-additive, epistatic interaction. To address this question, we made use of the fact that the TSS library included 342 variant pairs for which there were oligos representing all four combinations of the two alleles at the two variants. For each such variant pair, we tested whether the joint effect of both variants differed from the sum of their individual effects (*Figure 4—source data 1*).

There were five variant pairs that showed evidence for non-additive effects at an FDR of $\leq$ 20%. While the $\pi_1$ statistic (*Storey and Tibshirani, 2003*) suggested that at least 34% of variant pairs have non-additive effects, the majority of these cases could not be detected at individual significance, presumably due to limited statistical power.

The five significant pairs showed a range of epistatic patterns (*Figure 4* and *Figure 4—figure supplement 1*). For example, the RM alleles at two variants upstream of *DAD2* did not have significant effects individually (p>0.4 compared to the oligo with two BY alleles), but when combined, the two RM alleles drove significantly higher expression (p=2e-6, FDR = 0.01; *Figure 4*). RM carries a derived allele at each of these variants, indicated by the fact that the BY alleles are present in a strain from the Taiwanese clade of yeast isolates, which split early from other isolates (including BY and RM) during *S. cerevisiae* evolution (*Peter et al., 2018*). Both derived alleles are found at high and nearly identical frequencies in the yeast population (~74%), and the two variants, which are separated by only two nucleotides, show very high linkage disequilibrium (D'=0.999, $r^2$ = 0.99). Thus, these two epistatic variants form a tightly linked haplotype that increases gene expression only when both RM alleles are present, perhaps because both variants are needed to create a binding site for a transcriptional activator.

The RM alleles at two SNVs upstream of *RTT101* each reduced expression compared to the BY background (FDR < 0.1%), but an oligo carrying both RM alleles reduced expression to a similar degree as either variant alone (*Figure 4B*). Both RM alleles are derived, and the two variants have very little recombination (D'=0.998). However, the two RM alleles do not form a common haplotype ($r^2$ = 0.08), likely because while the RM 'A' allele at 352,322 bp on chromosome X is present in 42% of yeast isolates, the RM 'G' allele at 352,304 bp is much rarer, with a frequency of 6%. These results suggest that a common promoter variant reduces the expression of *RTT101*. A second, much rarer variant has a similar effect in isolation, but its effect is obscured in the presence of the more common variant, perhaps because both variants abrogate binding of the same transcriptional activator.

Together, these results show that *cis*-acting promoter variants can influence gene expression in a non-additive fashion. Such epistatic effects may be widespread, but the resultant deviations from additive expectations are often small, making their systematic detection challenging.

## Characteristics of causal variants

The identification of hundreds of causal variants enabled us to ask whether causal variants tend to share molecular or population genetic characteristics. To address this question, we assembled a set of 2998 features describing each variant. These features comprised sequence characteristics of the

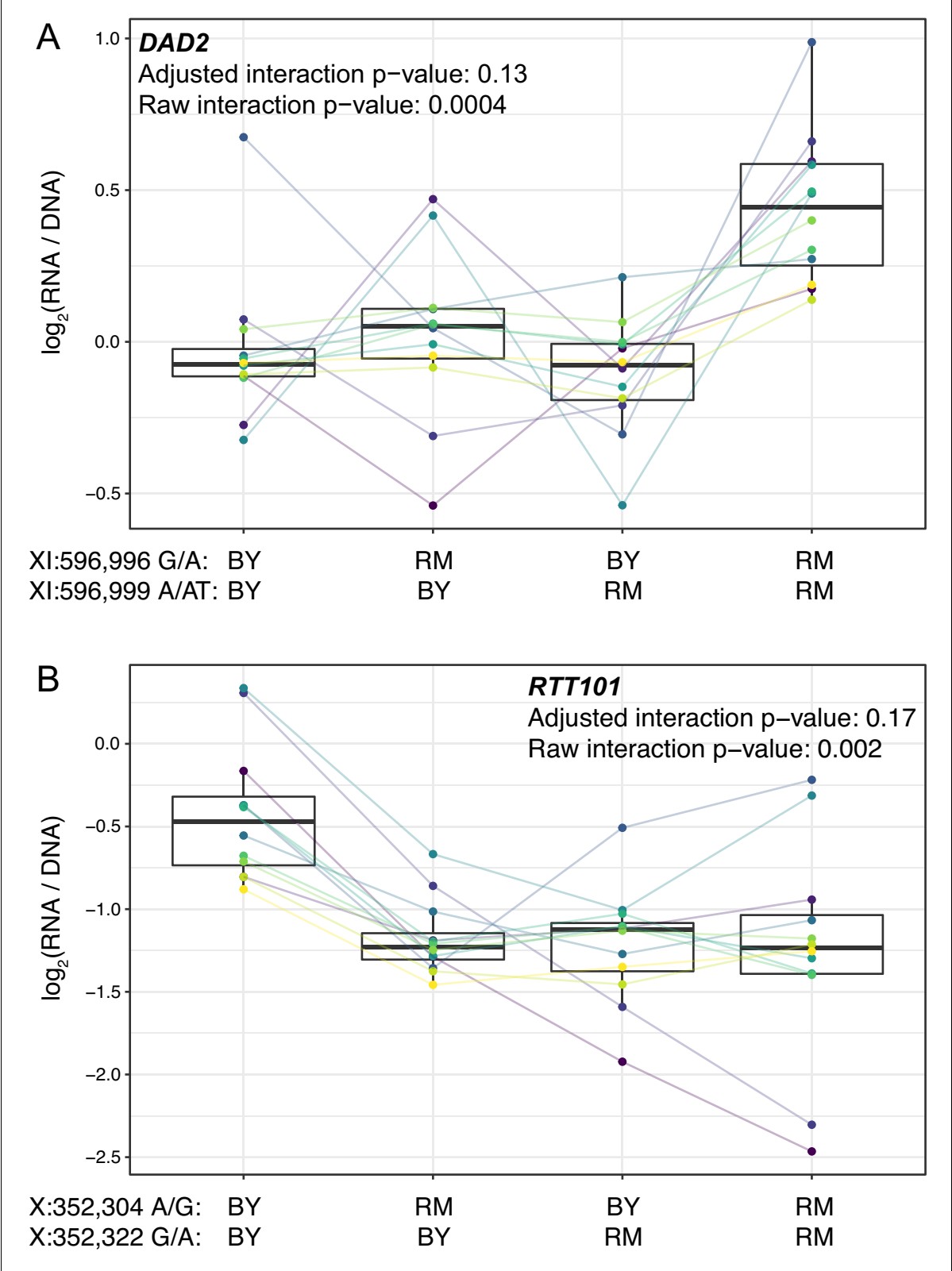

**Figure 4.** Epistasis among promoter variants. Each panel shows, for one gene, MPRA expression driven by four oligos with the indicated combination of BY and RM alleles at the two variants. Each panel states the gene name in bold along with multiple-testing adjusted and raw interaction p-values. Variants are given as "chromosome:position reference/alternative allele". Colored lines between boxplots connect the data for a given oligo in the different biological replicates. Boxplots show the median as thick line, with the box showing the 25th and 75th percentiles. Whiskers show the largest

*Figure 4 continued on next page*

*Figure 4 continued*

value no further than 1.5 times the inter-quartile range; any observations beyond this range are shown as individual points. (**A**). Promoter variants for DAD2. (**B**) Promoter variants for RTT101.

The online version of this article includes the following source data and figure supplement(s) for figure 4:

**Source data 1.** Results from the test for epistatic interactions.
**Figure supplement 1.** Additional cases of significant epistasis between promoter variants.

alleles, evolutionary properties of the variant, and features that describe the gene that is regulated by the promoter in which the variant resides (*Supplementary file 3* and *Supplementary file 4*). Most features (2,967) described transcription factor (TF) binding (see below).

We divided variants into a 'causal' group (defined as variants with an FDR of ≤5%; n = 459) and a non-causal group (variants with a raw p-value>0.2; n = 3,774). For variants located in divergent promoters that were assayed in both orientations, the results from the two orientations were considered separately. For this reason, the number of causal variants in these analyses was slightly higher than the 451 unique variants reported above. We used logistic regression to test the association of each feature with variant causality (*Figure 5—source data 1*). While these analyses cannot isolate the individual contributions of features that are correlated with each other, they provide an overview of the characteristics of causal variants.

Overall, causal variants were more likely to be SNVs than indels (*Figure 5A*). At the same time, longer indels were more likely to be causal than shorter indels (*Figure 5A*). Causal variants were less likely to be bound by a nucleosome in the genome (FDR = 4%). Nucleosomes may shield such variants from DNA-binding factors even on the plasmid reporter if the local DNA sequence around the tested variants can direct nucleosome formation at least partially.

Causal variants were enriched at nucleotides with higher PhastCons scores (FDR = 1%, *Figure 5A*), suggesting that natural variants that occur at nucleotides that are more conserved across yeast species are more likely to affect gene expression than those at less conserved nucleotides. Variants in the promoters of essential genes were less likely to be causal (FDR = 6%). Causal variants were also less likely to occur in promoters of genes with many synthetic genetic interactions (*Figure 5A*), which tends to be a property of essential genes (*Costanzo et al., 2016*). Essential genes are required for yeast viability, and are known to have less genetic (*Albert et al., 2018*), environmental (*Eng et al., 2010*), and stochastic (*Newman et al., 2006*) variation in gene expression. The observation that natural variants in promoters of essential genes are less likely to perturb gene expression is consistent with negative selection acting to purge causal variants from essential gene promoters. In further support of this hypothesis, causal variants had lower derived allele frequencies than non-causal variants (FDR = 10%), as expected if negative selection prevents variants that affect gene expression from rising to higher frequency (*Kita et al., 2017*; *Ronald and Akey, 2007*).

## Causal variants alter predicted transcription factor binding

Gene expression is shaped by chromatin state as well as by the binding of TFs to *cis*-regulatory elements (*Rando and Winston, 2012*). Previous work suggested that while *cis*-eQTLs can perturb multiple aspects of chromatin, these molecular changes tend to ultimately be caused by sequence-directed differences in TF binding (*Degner et al., 2012*; *Kasowski et al., 2013*; *Kilpinen et al., 2013*; *McVicker et al., 2013*). To explore the influence of single causal variants on TF binding, we computed the expected propensity of each of 196 TFs to bind to the two alleles at each variant. Because some TFs have different affinities for the two DNA strands at their binding sites (*de Boer et al., 2020*), we separately considered binding to the sense and the antisense strand, as well as in a strand-agnostic manner. TFs bind to individual 'strong' sites that closely match their motif, as well as to weaker sites with imperfect motif matches (*de Boer et al., 2020*). To capture this distinction, we computed separate feature sets that considered only strong or only weak binding sites. These analyses cannot disambiguate sharing of similar binding motifs by different TFs but probe the overall role of perturbed TF binding in variant causality. Finally, we also aggregated the 196 TF-specific feature sets into summary features that reported maximum and average allelic differences at each variant across the 196 TFs (Materials and methods).

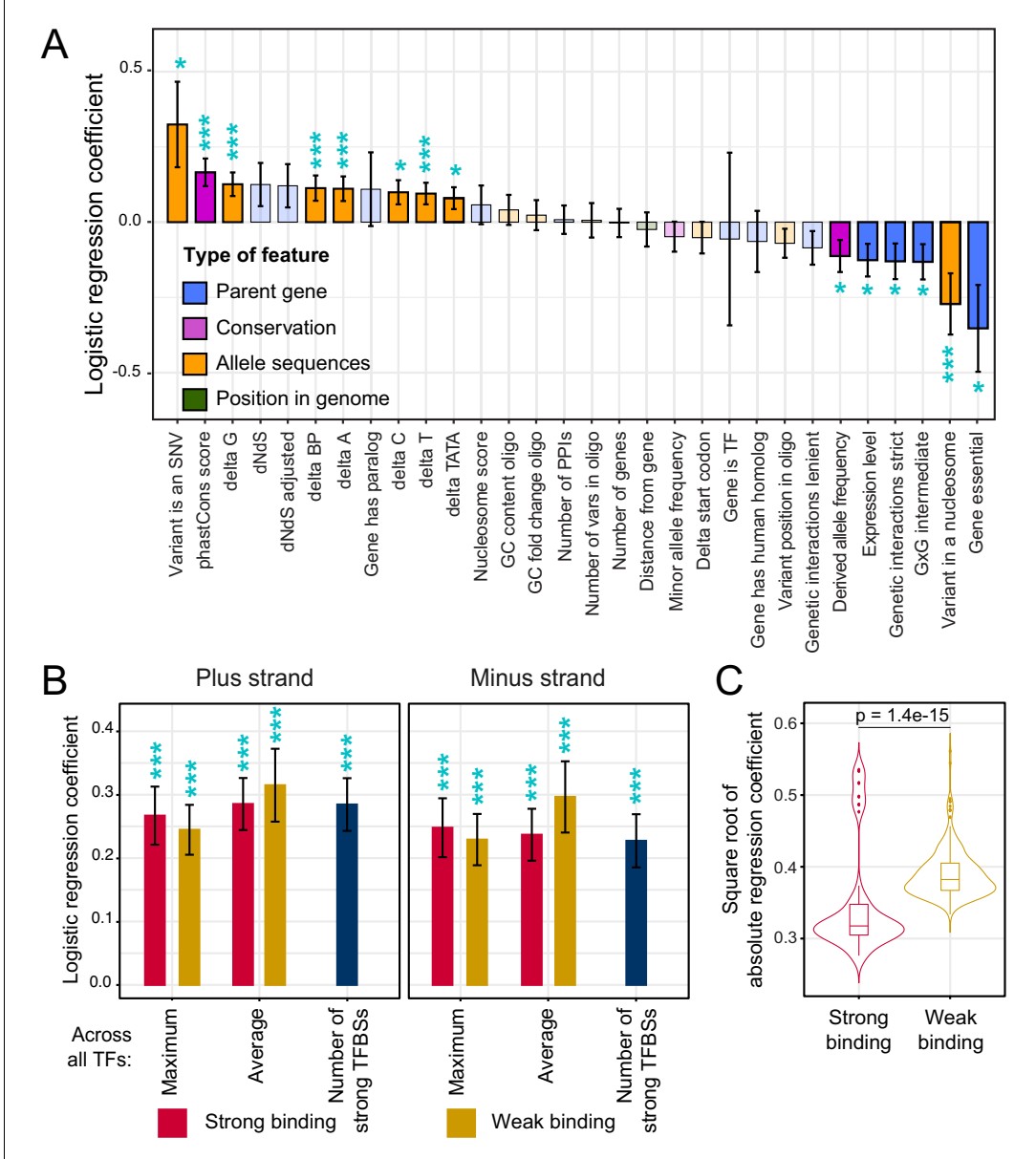

**Figure 5.** Association of features with variant causality. (**A**) Non-TF features. The figure shows the strength of association between each feature and variant causality. Error bars show the standard error of the mean. Significant associations are indicated by three stars (FDR < 5%) or one star (nominal p-value<0.05). Non-significant features are shown in lighter coloring. (**B**) TF summary features aggregated across all 196 TFs, separated by strand, mode of aggregation across TFs, strength of binding (weak or strong), and mode of comparing allelic PWM scores across sliding sequence windows spanning each variant (Materials and methods). Each of these summary features was significantly associated with variant causality at an FDR of <5%. (**C**) Distributions of logistic regression estimates for strong vs. weak binding for individual TFs. The p-value shows the result of a Wilcoxon rank test. See also *Figure 5—figure supplement 1*.

The online version of this article includes the following source data and figure supplement(s) for figure 5:

**Source data 1.** Single-feature regression analyses of causal variants.

**Figure supplement 1.** Examples of predicted TFBS changes at individual variants.

Overall, TF binding was strongly and broadly associated with variant causality (*Figure 5B*). Across the 2,940 TF-specific features, 481 features for 139 distinct TFs showed significant associations at an FDR of 5% (*Figure 5—source data 1*). All of the 27 summary features were significant at this threshold. For example, causal variants were more likely to result in gains or losses of strong binding sites

(62%; n = 286) than non-causal variants (49%; n = 1,846; FET p=6e-8). These associations were found on both DNA strands (*Figure 5B*).

Causality was associated with differences in strong as well as weak TF binding (*Figure 5B*; *Figure 5—figure supplement 1*). While yeast promoters harbor relatively few strong binding sites, they contain many binding sites that are individually weak but that in combination greatly influence promoter activity (*de Boer et al., 2020*). In our logistic regression analyses of the 196 individual TFs, differences in the various weak TF-binding metrics showed stronger associations with causality than those in the strong TF-binding metrics (*Figure 5C*). Because of the relatively low density of strong binding sites, many natural variants change no or only a small number of these sites. In contrast, the high density of weak binding sites presents a large mutational target, such that essentially every variant perturbs one or more weak binding sites. Evidently, these subtle differences can result in detectable expression changes. Overall, these analyses provided clear evidence that variants predicted to perturb TF binding are more likely to alter gene expression than variants not predicted to do so.

## Prediction of causal variants

The prediction of the effects of individual variants in a given genome is a major challenge for modern genetics and genomics. We tested whether the hundreds of causal variants we identified might enable us to predict the effects of individual *cis*-acting variants from their annotated features. We built 112 multiple logistic regression models for predicting variant causality from various feature subsets (*Figure 6—source data 1*). We trained these models using repeated 10-fold cross-validation on a random subset comprising 90% of the causal and non-causal variants and tested model performance on the remaining 10% of the variants. The best model achieved an area under the receiver operating curve (AUC) of 0.71 (*Figure 6A*; see inset for details on the model). Cross-validated elastic-net regularization, which is less prone to potential overfitting, did not improve model performance (best model AUC = 0.71).

We also built a series of multiple linear regression models that used the same 112 feature combinations to predict the absolute log-fold change caused by a given variant. When fit to the entire dataset, nearly all models fit the data better than expected by chance, with $r^2$ values of up to 0.33 at a p-value of 3e-50 for a model including non-TF features as well as all TF features (*Figure 6A*). A model considering only the TF features fit the data almost as well ($r^2$ = 0.32, p=7e-49). The median $r^2$ across the 112 models was 0.09. Next, we used 90% of the variants to train these linear regression models using repeated 10-fold cross-validation and predicted absolute log-fold change in the remaining 10% of the variants. The best correlation between predicted and observed fold-changes was rho = 0.15 (p=0.0004). Several other models performed nearly as well (*Figure 6A*; *Figure 6—source data 1*). The median model prediction performance was rho = 0.07. Models fit with cross-validated elastic-net regularization performed no better than the corresponding unregularized linear models (best rho = 0.12, p=0.02; median rho = 0.005). Overall, prediction models built from various feature sets were able to predict the identity and effect size of causal variants better than chance, but with modest accuracy.

Recently, de Boer et al. reported a model for predicting reporter gene expression driven by millions of random DNA fragments (*de Boer et al., 2020*). We tested whether this model was capable of predicting individual variant effects in our data. Using the DNA sequence centered on each variant in our library as input, we predicted reporter gene expression driven by each oligo in our libraries. These predicted expression values showed a significant correlation with measured expression in our MPRA (*Figure 6B*). We then computed the predicted fold change of each variant as the difference between the predicted expression for the respective BY and RM oligos. The predicted and measured fold-changes of all variants correlated significantly (*Figure 6C*). This correlation increased for variants that were causal in our data (*Figure 6C*), although this subset would be unknown during variant effect prediction from genome sequence alone. When we ignored the direction of the fold-changes, prediction performance was reduced for all variants (rho = 0.09, p=1e-8), and degraded completely for causal variants (rho = 0, p=0.9). This suggests that much of the ability of this independent model to predict variant effects in our data derives from the model's ability to correctly capture whether altered binding of individual transcription factors to a given allele increases or decreases expression.

In summary, models based on individual features trained on our variant effects, as well as a completely independent model (*de Boer et al., 2020*), showed similar ability to predict variant

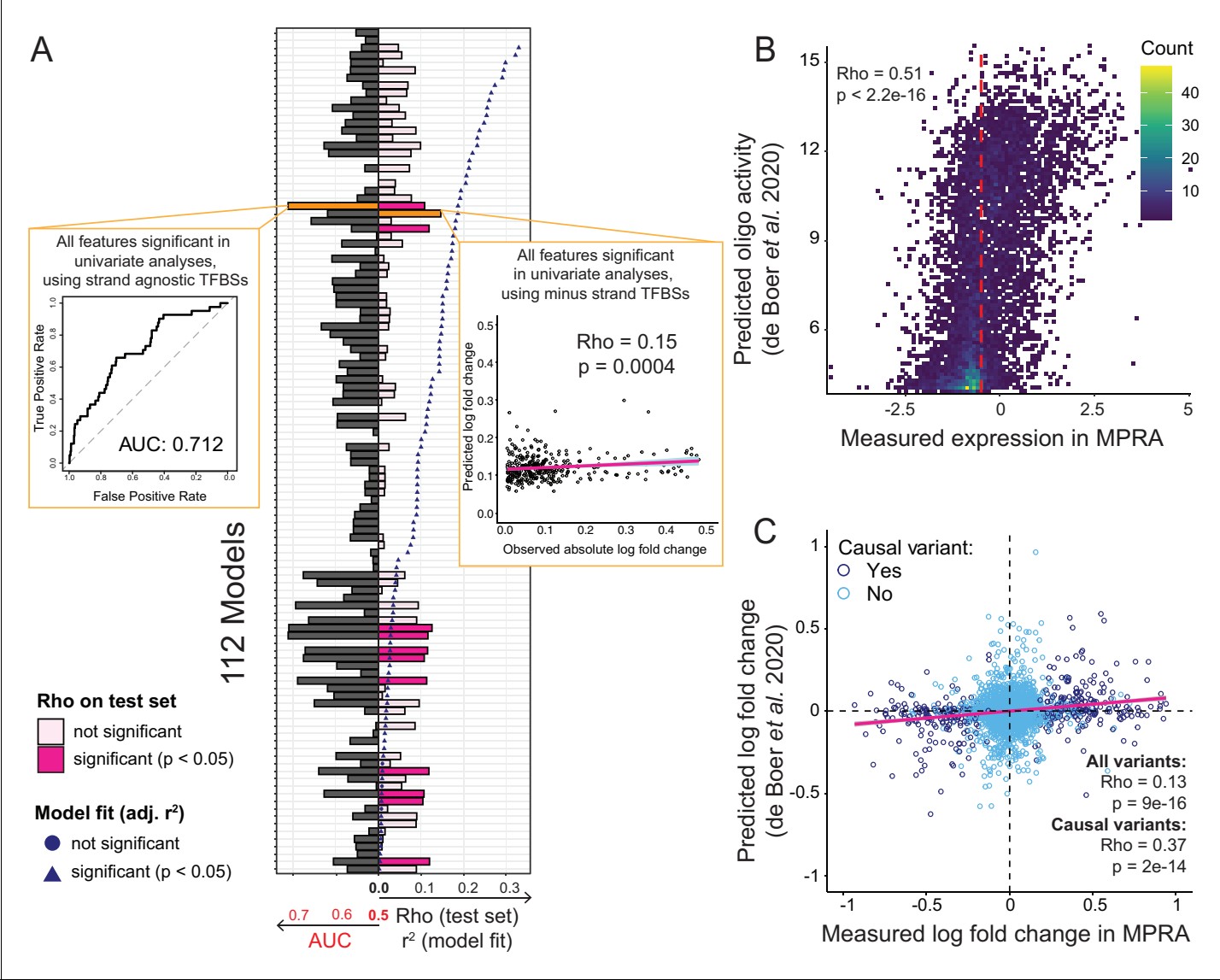

**Figure 6.** Prediction of causal variants and variant effects. (A) Prediction results for 112 models. On the left, the plot shows the performance of binomial classifiers on the 10% test as black bars. On the right, the plot shows the performance of the linear predictors of variant effects as spearman correlation coefficients (rho) between actual and predicted log-fold changes in expression as pink bars. Blue symbols show r2 values for each model when fit to the entire dataset. The best classifier and linear model are indicated by orange bars and shown as insets. (B) Measured expression driven by each oligo versus expression predicted by the de Boer model. The dashed red line denotes the median measured expression level. (C) Observed fold-changes of individual variants measured in our MPRA versus fold-changes predicted by the de Boer model. The pink line shows the linear regression fit for all variants.

The online version of this article includes the following source data for figure 6:

**Source data 1.** Multiple-feature regression analyses of causal variants.

effects from DNA sequence. Although prediction performance was modest in both cases, the fact that a model trained on an independent experiment can capture some effects in our data demonstrates that the prediction of individual variant effects from DNA sequence alone is becoming possible.

## Discussion

In this work, we used an MPRA to examine the effects of roughly half of all intergenic DNA sequence variants between two yeast strains and identified hundreds of variants with significant, *cis*-acting

effects on gene expression. These significant variants reflected the effects of known local eQTLs, suggesting that we have identified causal variants that underlie natural regulatory variation. Several insights emerged from in-depth analysis of these causal variants.

First, the molecular basis of local eQTLs can include single (as for *OLE1*) or multiple (as for *CWP1*) causal variants. Studies seeking to identify the molecular basis of QTL regions often implicitly assume that each QTL is driven by a single causal variant. However, efforts to fine-map QTLs for organismal traits often find that multiple, linked causal variants exist in a region (*Kroymann and Mitchell-Olds, 2005*; *Sinha et al., 2008*; *Steinmetz et al., 2002*). Whether multiple variants also underlie QTLs for molecular traits such as gene expression was less clear, especially for local eQTLs in yeast, given the small mutational target size of the compact regulatory regions. Our results show that the molecular basis of local regulatory variation can involve multiple causal variants even in a single pair of yeast strains.

In some promoter regions (as for *UFO1*), we detected variants with effects in opposing directions that cancelled each other out, resulting in no local eQTL signal, demonstrating that QTL mapping can miss linked causal variants of opposite effect. Similarly, recent work has shown that the number of *trans*-acting loci affecting gene expression throughout the genome may have been greatly underestimated due to linkage in experimental crosses with limited recombination (*Metzger and Wittkopp, 2019*). These results paint an emerging picture of a highly polygenic architecture of gene expression variation, with dozens of *trans*-acting loci across the genome (*Albert et al., 2018*; *Albert et al., 2014b*; *Brion et al., 2020*; *Metzger and Wittkopp, 2019*), each of which may harbor multiple causal variants.

Second, we detected several instances of non-additive epistatic interactions between variants in the same promoter. Previous work revealed epistatic eQTL pairs that influence the expression of a given gene, including between *cis* and *trans* eQTLs as well as between *trans* eQTLs (*Albert et al., 2018*; *Brem et al., 2005*). Very few of these cases have been resolved to individual variants (*Brem et al., 2005*; *Lutz et al., 2019*). Our results show that there can also be epistasis between multiple causal variants within a single local eQTL. Nevertheless, most of our assayed variant pairs act additively. Even for pairs with the most significant epistasis, the quantitative departures from additivity tended to be small (*Figure 4* &S10). This result is consistent with searches for epistatic QTLs for various growth traits in this cross. Although examples of higher-order epistatic interactions with profound effects on some traits have been reported (*Forsberg et al., 2017*), most of the hundreds of detected epistatic pairs involve small deviations from additivity that account for minuscule fractions of phenotypic variance (*Albert et al., 2018*; *Bloom et al., 2015*), as expected from quantitative genetic theory (*Hill et al., 2008*).

Third, *cis*-acting variants showed signals of evolving under negative selection, in that the derived alleles of causal variants tended to have reduced population frequencies and were less likely to be found in the promoters of essential genes. Similar results reviewed in *Signor and Nuzhdin, 2018* have been reported in yeast (*Kita et al., 2017*; *Ronald and Akey, 2007*), plants (*Josephs et al., 2015*) and humans (*Battle et al., 2014*), but were based on studies in crosses subject to linkage among neighboring variants, or in natural populations subject to both linkage disequilibrium and differences in statistical power for variants with different population frequencies. Our MPRA results are free from confounding effects of linkage and population frequency and extend the observation of negative selection to single, causal *cis*-acting variants.

Fourth, causal variants were enriched for sequence changes that altered predicted transcription factor binding. This enrichment was present for strong binding sites, as well as weak sites that do not pass strict binding site detection thresholds. Emerging evidence suggests that in addition to strong, high-affinity binding sites, intergenic DNA harbors an abundance of weak, low-affinity binding sites with imperfect motif matches (*de Boer et al., 2020*; *Tanay, 2006*; *Wunderlich and Mirny, 2009*). This latter group provides a larger mutational target than the few strong exact motif matches. As a consequence, almost every natural variant may perturb one or multiple weak binding sites. The joint effects of perturbations to weak sites could further contribute to molecular architectures in which several causal variants shape the overall activity of a given promoter in a given individual. While more work is needed to explore the consequences of alterations to weak binding sites, they may also help explain why many inferred causal variants in human diseases are located close to, but do not obviously change, recognizable TF motifs (*Farh et al., 2015*).

Our approach had several limitations. First, not all potential *cis*-acting variants were assayed in our libraries, including roughly half of intergenic variants as well as variants in gene regions such as the most proximal bases of the core promoter, the gene body, the 3'UTR, and the terminator. Future work with dedicated libraries to study these variants will likely reveal many additional causal effects. Meanwhile, the concordance of our results with local eQTLs suggests that we have discovered a non-trivial fraction of causal variants, and that, by extension, many *cis*-acting variants do indeed reside in promoter regions. Second, causal variants typically have small effects on gene expression, necessitating replicate assays to ensure high statistical power. Higher levels of replication, as well as further optimization of assay design, are likely to improve variant detection and effect quantification. Finally, our plasmid-bound assay cannot capture the full chromatin state of variants in the genome. Nevertheless, our causal variants correlated with local eQTL effects measured in the native genomic context, showing that chromatin state does not override the effects of all variants. This agreement was stronger for variants that reside in nucleosome-free regions than for nucleosome-bound variants. If nucleosomes do not form on the plasmids, such regulatory variants may be exposed to a regulatory environment similar to that in the genome. More broadly, the precise chromatin context in which a variant is embedded likely influences its exact effect on gene expression, consistent with the considerable differences revealed by MPRAs inserted at different sites in the human genome (*Maricque et al., 2019*), as well as with the high degree of tissue-specificity of human *cis*-eQTLs (*GTEx Consortium et al., 2017*).

The prediction of the consequences of individual DNA variants is a major area of research, especially given that accurate predictions could aid discovery of disease-causing mechanisms and diagnosis of individual patients with unknown genetic disorders. As a consequence, a plethora of computational methods aiming to predict the severity or molecular consequences of individual variants has been developed (for example, *Huang et al., 2017*; *Kircher et al., 2014*; *Lee et al., 2015*; *Zhou et al., 2018*; *Zhou and Troyanskaya, 2015*, reviewed in *Nishizaki and Boyle, 2017*). Although our models integrating various features to predict the identity and effect size of causal variants performed better than chance, no single feature and no simple combination of features achieved high prediction accuracy. While the size of our dataset, which included several hundred causal variants, may be insufficient to train predictors with better performance, an independent model trained on millions of DNA sequences (*de Boer et al., 2020*) and applied to our data performed similarly. Our prediction accuracies are also comparable to those obtained from state-of-the-art machine-learning tools applied to human MPRA data (*Kircher et al., 2019*) and to simulated genetic architectures (*Liu et al., 2019*). Clearly, prediction of variant effects remains a challenging problem even in comparatively simple yeast promoters. Models trained in a range of environmental conditions, which would provide variation in chromatin contexts, as well as on MPRA data with longer and more diverse flanking sequences (*Kircher et al., 2019*) may further improve predictive accuracy.

## Materials and methods

**Key resources table**

| Reagent type (species) or resource | Designation | Source or reference | Identifiers | Additional information |
|---|---|---|---|---|
| Strain, strain background (*Saccharomyces cerevisiae*) | BY4741 | *Albert et al., 2018* (doi:10.7554/eLife.35471) | | |
| Recombinant DNA reagent | RCP83 plasmid | This paper; (Addgene plasmid #163466) | | Plasmid backbone |
| Recombinant DNA reagent | SurePrint Oligonucleotide Libraries | Agilent | | Custom DNA oligo library |
| Gene (*Aequorea victoria*) | yEGFP | pKT0127 (Addgene plasmid #8728) | | Reporter gene |
| Software, algorithm | R version 3.5.0 | https://www.r-project.org | | Data analysis |

MPRA design and analysis code generated for this paper is available at https://github.com/frank-walbert/promoterVariants (copy archived at swh:1:rev:fb7e232981f63281d944ccf273fdafa24ac2272d; *Albert, 2020*). Unless otherwise specified, analyses were performed in R (https://www.r-project.org), using various packages from the tidyverse (https://tidyverse.tidyverse.org/index.html) (*Wickham et al., 2019*) and Bioconductor (https://www.bioconductor.org) (*Huber et al., 2015*). Specific packages are listed below. Sequences of primer and elements of the reporter construct are available in *Supplementary file 5*.

## MPRA library design

Our design was based on 45,543 variants previously detected from short-read sequencing data of the BY and RM strains (*Bloom et al., 2013*). Because the BY strain is nearly identical to the genome reference strain (*Engel et al., 2014*), RM alleles are by definition 'alternative' alleles. Variants close to the telomeres were excluded as described in *Albert et al., 2014b*. Variants included SNVs and short indels (*Supplementary file 6*). When multiple alternative alleles were reported for a variant, we used the alternative allele with the highest likelihood. We obtained gene annotations for the sacCer3 yeast reference genome build from SGD (https://www.yeastgenome.org) (*Cherry et al., 2012*). Coding genes, as well as tRNAs, snoRNAs, snRNAs, and other non-coding RNAs were included in the design. Noncoding genes that overlapped a coding gene were removed, along with genes on the mitochondrial and 2μ plasmid genomes.

We defined transcription start sites based on full-length yeast transcript isoforms obtained by capturing and sequencing both the 5' cap and 3' polyadenylation site of yeast RNAs (*Pelechano et al., 2013*). Specifically, we considered transcript isoforms reported in Supplementary Data S2 from *Pelechano et al., 2013* that completely contained their annotated gene, avoiding isoforms that initiated transcription inside of annotated gene models or that terminated transcription prematurely. For each gene, we summed the number of reported counts for all isoforms that started at the same 5' base in YPD medium. Based on these summed counts, we selected the 5' position with the most read counts to represent the transcription start site for the given gene.

For the TSS library, we included DNA variants between BY and RM located within the 144 bp upstream of the transcription start site for each gene. We designed 'oligo blocks', sets of oligos carrying allelic versions of these 144 bp sequences. We extracted the reference genome sequence for these regions to form oligos carrying BY alleles at all variants. For each variant in the 144 bp, we created one additional oligo carrying the RM allele at this variant. We also generated one oligo per block that carried RM alleles at all variants. The TSS library comprised such oligo blocks for all genes that harbored variants in the design space of this library.

For the Upstream library, we considered all variants located between 72 bp upstream of the start codon of each gene and the coding sequence of the next upstream gene, up to a maximum distance of 1 kb. For each such variant, we created an oligo block consisting of one oligo that carried the BY allele at the variant flanked by the reference genome sequence centered on the variant and one oligo that carried the same flanking sequence but with the RM allele at the variant. At any other variants within the sequence covered by the oligo block, both oligos carried the BY allele such that each oligo block assayed a single variant. We subsampled 4547 variants from this design. Specifically, we included all variants located in promoters whose genes had at least some evidence (nominal p<0.05) of allele-specific expression (ASE) in either of two ASE datasets (*Albert et al., 2018*; *Albert et al., 2014a*). We also included variants from 1000 randomly selected genes without evidence of ASE (p>0.2), as well as 2062 additional variants selected at random from the remaining variants.

Both designs were strand-specific, such that for genes located on the plus strand we designed oligos based on the reference genome sequence, while for genes located on the minus strand, we used the reverse-complement of the reference sequence. Thus, in the finished reporter constructs, all oligos were correctly oriented with respect to the fluorescent reporter gene.

We removed oligo blocks in which at least one oligo happened to contain a restriction site that would be recognized by the restriction enzymes we used during cloning. We also removed oligo blocks in which RM insertion variants increased the length of at least one oligo to more than the synthesis limit of 200 bp.

In both libraries, we included an identical set of 200 oligos based on promoter fragments studied by *Sharon et al., 2012*. Because the promoter fragments in that study were shorter (103 bp) than

ours (144 bp), we added a random DNA sequence (TATAGAACGGAATCACCTCTGACAAGTAGCG TCAAATCGGT) between the AscI cloning site and the p2 priming site such that this random sequence was removed during cloning (*Figure 1—figure supplement 3*). This extra sequence safeguarded against preferential PCR amplification of what would otherwise have been shorter oligo molecules.

To each designed oligo sequence, we added restriction sites and library-specific priming sites as shown in *Figure 1—figure supplement 3*. The resulting oligos had a median length of 195 bp. RM insertion variants increased the length of some oligos, such that the maximum oligo length was 200 bp. Because oligo synthesis is challenging for sequences rich in adenine (which are enriched in promoter sequences), we reverse-complemented the designed sequences of oligos that carried more adenines than thymines. Because the single-stranded oligo libraries are amplified by PCR prior to use, these reverse-complement oligos behaved equivalently equivalent to the designed oligos in downstream experiments. Oligos were purchased as one pool from Agilent Technologies.

## Reporter design

Inspired by the design of prior work (*Sharon et al., 2012*), we placed the promoter fragment library upstream of a fluorescent yEGFP gene on a plasmid (*Figure 1—figure supplement 3*). Similar to *Kosuri et al., 2013*, we measured reporter gene expression by quantifying yEGFP mRNA instead of fluorescence signal. Plasmids carried a CEN/ARS origin of replication and an *ADH1* terminator downstream of the reporter constructs. We included random 20 bp barcodes downstream of the yEGFP gene, such that the barcodes were transcribed as part of the yEGFP mRNA. Prior to insertion of the yEGFP gene into plasmids, we used paired-end sequencing to generate a dictionary linking each random barcode to the promoter fragment it tagged. We amplified the barcode from plasmid DNA or yEGFP cDNA to generate Illumina sequencing libraries. Reporter expression was quantified by high-throughput sequencing and counting of barcode reads. Details on these procedures are given below.

For the TSS library, the designed promoter fragments were inserted immediately upstream of the yEGFP ORF. For the Upstream library, an invariant *HIS3* minimal promoter was inserted between the oligo sequences and the yEGFP ORF (*Sharon et al., 2012*).

## Initial library amplification and barcoding

The TSS and Upstream libraries were selectively amplified from the common oligo pool (Agilent). Per library, each of eight parallel reactions contained 0.2 µL of oligo pool diluted 1:10 in TE buffer for ~10 ng template DNA, 5 µL of 2x Kapa library amplification mix (Roche, KK2702), 0.2 µL forward primer (10 µM; TSS: orc281, Upstream: orc279), 0.2 µL reverse primer (10 µM; TSS: orc285; Upstream: orc283), and 4.4 µL water. Cycling conditions were: 98 ˚C for 2 min, 5 cycles of 98 ˚C for 3 s, 50 ˚C for 20 s, 60 ˚C for 10 s, 10 cycles of 98 ˚C for 3 s, 60 ˚C for 30 s, and a final extension at 60 ˚C for 2 min followed by a hold at 10 ˚C. The eight parallel reactions were pooled, cleaned up using QiaQuick PCR purification kits (Qiagen #28106), eluted in 30 µL EB buffer, and quantified with Qubit HS DNA kits (Thermo Fisher #Q32854).

To generate barcoded libraries, we performed PCR in eight parallel reactions. Each contained 10 ng of amplified library, 10 µL 5x Phusion buffer, 1 µL dNTP mix (10 mM, Invitrogen #18427013), 1 µL forward primer (10 µM; TSS: orc291, Upstream: orc287), 1 µL reverse primer (10 µM; TSS: orc292, Upstream: orc288), 0.5 µL Phusion HS II DNA Polymerase (ThermoFisher #F549L), and water to 50 µL. Cycling was performed as follows: 98 ˚C for 2 min, 14 cycles (TSS) or 12 cycles (Upstream) of 98 ˚C for 3 s, 60 ˚C for 30 s, 72 ˚C for 30 s, a final extension at 72 ˚C for 2 min, and a hold at 10 ˚C. The eight parallel reactions per library were combined, cleaned up using QiaQuick PCR purification kits, eluted in 30 µL EB buffer, and quantified with Qubit HS DNA kits.

Because the barcodes are part of the reporter mRNA molecules, their sequence potentially influences mRNA abundance in *cis*, for example by affecting mRNA stability. Indeed, pilot experiments suggested that barcodes with a high number of guanines had lower expression, in line with known effects of guanines in the 3'UTR (*Shalem et al., 2015*). Barcodes starting with two guanines had particularly low expression, and an adenine at the first position increased expression variance. To increase overall library expression while maintaining high barcode complexity, all experiments reported here used random barcodes in which (1) the first base was either cytosine or thymine, (2)

the next three positions and every remaining even position contained no guanines but each of the other three DNA bases at equal frequency, and (3) the remaining odd positions carried each of the four bases at equal frequency.

## Plasmid library generation

The base plasmid 'RCP83' was generated synthetically (Gen9/Ginkgo Bioworks) with a bacterial ampicillin marker and a yeast kanMX resistance cassette (*Supplementary file 7* contains the map of this plasmid after completed library construction; the RCP83 backbone is available at Addgene under ID #163466). The barcoded library was directionally ligated into the plasmid at an SfiI site. Upstream of this site, the plasmid contained the 'PS1' priming site used in barcode annotation (see below). Downstream of the SfiI site, the plasmid carried an *ADH1* terminator, ensuring efficient termination of the reporter constructs. We isolated RCP83 from *E. coli* cultures by maxi-prep using Qiagen Plasmid Plus Purification kits (Qiagen #12963, #12965) and quantified plasmid DNA using Qubit dsDNA HS Assay Kit (Thermo Fisher #Q32854). We digested 10 µg of RCP83 using 10 µL SfiI enzyme (NEB, #R0123S) in 50 µL 10x NEB CutSmart buffer and water to 500 µL. This mixture was incubated at 50 °C for 4 hr and cooled to room temperature. To prevent religation of the backbone, we added 10 µL of Shrimp Alkaline Phosphatase (NEB #M0371L) and incubated at 37 °C for 1 hr. We purified the digested vector from a 0.8% agarose gel (1.2 g agarose Fisher #21-255-00GM) in 150 mL 1x TAE (Fisher #FERB49) using a QiaQuick Gel Extraction Kit (Qiagen #28706) and quantified using a Qubit dsDNA HS Assay Kit.

We digested 1 µg of barcoded library using 5 µL SfiI enzyme, 10 µL 10x NEB CutSmart buffer, and water to 100 µL at 50 °C for 2 hr. Digested insert was purified using QiaQuick PCR purification and eluted in 30 µL EB.

Ligation of the digested library into the digested backbone was performed by mixing 1 µg of vector, 100 ng of library, 80 µL of 10x T4 DNA ligase buffer, 4 µL of T4 DNA ligase (NEB #M0202M), and water to 800 µL. The mixture was incubated at 16 °C for 18 hr, 65 °C for 20 min, and held at 12 °C. The reaction was cleaned up with a DNA Clean and Concentrator kit (Zymo Research, #D4013) and eluted in 25 µL water.

Transformation into *E. coli* was performed by electroporation into *E. cloni* 10G SUPREME Electrocompetent Cells (Lucigen, #60080–2), in 11 parallel transformations (1 mL each) on a Bio-Rad MicroPulser (Bio-Rad, #165–2100). The parallel transformations were combined and mixed with a total of 9.5 mL of recovery medium. To estimate the number of transformants obtained, we plated a dilution series (10-fold dilution steps from 1:100 to $1:10^6$) on LB carbenicillin (100 µg / mL) plates, grew the plates overnight at 30 °C and counted colonies after 24 hr. We obtained an approximate of 25 million and 230 million transformants for the TSS and Upstream libraries, respectively. Negative controls, for which we had performed the ligation and transformation with digested backbone but no insert, yielded negligible numbers of colonies. The transformed cells were plated on 15 cm LB + carbenicillin plates (500 µL per plate), grown overnight at 30 °C, and scraped into a total of 25 mL LB medium. The optical density ($OD_{600}$) of these cells suspensions was determined, and 2 $OD_{600}$ units (~1 billion cells) were grown overnight at 30 °C in 400 mL LB + carbenicillin. After 24 hr, plasmids were isolated from 200 mL of these cultures using Qiagen Plasmid Plus Purification kits (Qiagen #12963, #12965). The remaining 200 mL of the overnight cultures were spun down, resuspended in 6 mL 20% glycerol, and frozen at −80 °C in 1 mL ultra-concentrated aliquots.

## Barcode annotation

We annotated designed promoters to random barcodes by high-throughput sequencing. To generate the sequencing libraries, we performed four parallel PCR reactions each for the TSS and the Upstream library. In each reaction, we combined 800 ng of plasmid library with 25 µL 2x Kapa library amplification mix (Roche, #KK2702), 1 µL forward primer (10 µM; orc295), 1 µL reverse primer (10 µM; orc296), and water to 50 µL. PCR was performed as follows: 98 °C for 2 min, 10 cycles of 98 °C for 3 s, 72 °C for 30 s, and a final extension at 72 °C for 2 min with a hold at 12 °C. The four reactions were combined, and clean-up performed using 360 µL AmPure XP beads for 200 µL combined reactions. Beads were incubated with the library at room temperature for 5 min, separated on a magnetic rack, washed twice with 500 µL 80% EtOH, and eluted from the beads with 50 µL elution (TE)

buffer for 5 min. Sequencing libraries were quantified using qPCR Library Quantification kits (Roche, #KK4824).

We sequenced these annotation libraries in paired-end configuration with 250 bp read length on two lanes of an Illumina HiSeq 2500 instrument. At this read length, both of the paired reads covered the barcode and the associated oligo entirely. This configuration aided in distinguishing sequencing errors, which are present on only one of the paired reads, from synthesis and PCR errors in the sequenced molecule and are therefore present in both paired reads. Paired reads were merged using PEAR (*Zhang et al., 2014*) using parameters -v 75 m 250 j 8 -y 2G -c 40 -b 64. Between 75 and 85% of read pairs were merged successfully.

We retained merged reads that contained two invariant sequences expected to be present in well-formed sequencing reads: in between the oligo and the barcode (TSS: CCTGCAGGGG TTTAGCCGGCGTG; Upstream: CCTGCAGGGTTCCGCAGCCACAC; these correspond to the AscI, p2, and SbfI sequences; *Figure 1—figure supplement 3*, *Supplementary file 5*) and upstream of the oligo (GGCCGTAATGGCC in both libraries; corresponding to the SfiI-A site included in the synthetic oligo sequences; *Figure 1—figure supplement 3*, *Supplementary file 5*). These invariant sequences were used to locate the positions of the barcode and oligo in each read. This filter retained ~ 170 million reads in the TSS library and ~125 million reads in the Upstream library, representing 24 million (TSS) and 46 million (Upstream) unique barcodes. Most of these unique barcodes were rare (*Figure 1—figure supplement 4*), as indicated by median read counts per barcode of two (TSS) and one (Upstream). We did not group barcodes with similar sequences at this stage.

We examined barcodes that appeared to tag multiple oligos and concluded that the overwhelming majority of barcodes tagged a single oligo. Specifically, for the overwhelming majority of cases apparent tagging of multiple oligos, a single 'primary' oligo dominated the barcode, usually making up at least 90% of oligo reads for the barcode. Examination of the 'secondary' oligos showed that these usually differed from the primary oligo by a single nucleotide, most often involving a skipped base. Because our paired-end sequencing strategy reduced sequencing errors, we deemed it likely that these secondary oligos arose as PCR errors during barcoding or Illumina sequencing library generation. In downstream analyses, we only considered the primary oligo as the oligo tagged by the given barcode.

We next mapped these primary oligos to our designed sequences. Synthesis and PCR errors result in oligos that do not match the designed sequences perfectly. Because our interest was in the effects of individual sequence variants that usually alter just a single nucleotide, errors in the oligo molecules run the risk of dominating the modest effects exerted by natural regulatory variants. Therefore, we retained only barcodes for which the primary oligo perfectly matched a designed oligo. In the TSS library, we retained 9.2 million barcodes (38% of unique barcodes), while in the Upstream library we retained 20 million barcodes (43%).

In the TSS library, these barcodes tagged 6565 oligos, which was 91% of all designed TSS oligos. In the Upstream library, the barcodes tagged 9646 oligos, representing 98% of the design. Examination of designed oligos that were not present in the cloned libraries revealed that oligos starting with a guanine had lower representation in the library (*Figure 1—figure supplement 5*), especially for oligos that started with two guanines. This technical bias was more apparent in the TSS library, in which a higher fraction of oligos started with disfavored nucleotides than in the Upstream library.

Raw data and barcode assignments to oligos are available under GEO accession GSE155944 (https://www.ncbi.nlm.nih.gov/geo/query/acc.cgi?acc=GSE155944).

## Reporter subcloning

After plasmid library generation, the fluorescent reporter gene was subcloned in between the promoter fragments and the barcodes of the annotated plasmid libraries by restriction digestion and ligation. Specifically, we used a codon-optimized yEGFP based on sequence from plasmid pKT0127, a gift from Kurt Thorn (Addgene plasmid #8728; https://www.addgene.org/8728/). In the Upstream library, the 100 bases upstream of the *HIS3* gene (*Supplementary file 5*) were included as an invariant minimal promoter between the oligo library and yEGFP, as in *Sharon et al., 2012*. yEGFP and the *HIS3* minimal promoter sequences were synthesized by Genewiz and were amplified using primers orc362_yeGFP (TSS) or orc362_HIS3 (Upstream), and orc363. These sequences were subcloned into the base plasmid (RCP83) to generate the final reporter plasmids.

Reporter gene inserts and the barcode-mapped plasmid library were digested as follows. We mixed 20 µg of insert or 15 µg plasmid library with 5 µL AscI (NEB R0558L), 5 µL SbfI-HF (NEB R3642L), 50 µL 10x CutSmart NEB buffer, and water to 500 µL. These mixtures were incubated at room temperature for 2 hr (insert) or 4 hr (plasmid library). The library plasmids were cooled to room temperature. We added 10 µL rSAP (NEB #M0371L) and incubated at 37 °C for 1 hr to prevent plasmid re-ligation. Digests were gel purified on 2% (insert) agarose or 0.8% (plasmid library), and bands at 700 bp (insert) or 6 kb (plasmid library) cut and purified using QiaQuick Gel Extraction Kit (Qiagen #28706) with elution into 30 µL EB.

Ligation of the digested reporter gene into the digested library plasmids was performed by mixing 2 µg (TSS) or 1 µg (Upstream) of vector, 700 ng (TSS) or 233 ng (Upstream) of insert, 80 µL of 10x T4 DNA ligase buffer, 8 µL of T4 DNA ligase (NEB #M0202M), and water to 800 µL. The mixture was incubated at 16 °C for 18 hr, 65 °C for 20 min, and held at 12 °C. The reaction was cleaned up and eluted in 25 µL water using the Zymo DNA clean and concentrator kit.

The ligated reactions were transformed into *E. coli* as described above for first step cloning, with 11 parallel transformations per library. Yields were 33 million (TSS) and 1.9 million (Upstream) cells per mL. Per library, 10 mL transformation mix was plated on twenty 15 cm LB + carbenicillin plates, grown overnight at 30 °C, and harvested by scraping in 5 mL LB per plate. From the resulting dense cell mix, we grew 2 billion cells in 100 mL LB + carbenicillin medium at 30 °C for ~ 20 hr for plasmid isolation. We condensed 45 mL of cells into 4 mL aliquots of 30% glycerol.

For the TSS library, we grew one aliquot overnight in 400 mL LB + carbenicillin medium at 30 °C and performed three maxi preps and two mega preps (200 mL each) using Qiagen Plasmid Plus Purification kits (Qiagen #12963, #12965, #12981). For the Upstream library, we sent glycerol stocks for large-scale plasmid prep (Genewiz) yielding 491 µg of plasmid library.

Library integrity was confirmed by restriction analysis with AscI and SbfI-HF. We also created next-generation sequencing libraries targeting the barcodes as described below and sequenced them on an Illumina MiSeq instrument to confirm that the libraries had retained high barcode complexity after subcloning.

## Yeast strain and media

We used a prototrophic yeast laboratory strain BY (*MATa*) without resistance cassettes (YLK1879) (*Bloom et al., 2013*).

YNB (MSG) + glucose + G418 medium was prepared as follows: dissolve 6.7 g YNB without amino acids and without $NH_4SO_4$ and 1 g monosodium glutamate in 900 mL $H_2O$, autoclave, let cool to room temperature, add 100 mL sterile glucose (20%), add 1 mL G418 (1,000X).

YPD medium was made by combining 5 g yeast extract (Fisher #212720), 10 g peptone (Fisher #211820), and 435 mL milliQ water, autoclaving, and adding 50 mL 20% glucose (Fisher #901521).

## Yeast transformation

The libraries were transformed into using the LiAc method (*Gietz and Schiestl, 2007*). For the TSS library, we transformed 1.6 L of yeast growing in YPD at an $OD_{600}$ of 1.25 with 248 µg plasmid DNA. We split the 1.6 L culture in half, harvested by centrifugation at 3,000 g for 5 min at 20 °C, washed twice with water (once with half and once with 1/5 of the culture volume), and pelleted by centrifugation. Each of the two cell pellets was mixed with transformation mix (28.8 mL PEG 50% w/v, 4.32 mL LiAc 1M, 6 mL single-stranded carrier DNA at 2 mg / mL, water to 43.2 mL, as well as half the plasmid DNA) in a 50 mL conical tube. Prior to making the transformation mix, single-stranded carrier DNA was denatured for 5 min in a boiling water bath and chilled on watery ice. Transformation mix was kept on ice until use. Cells mixed with transformation mix were heat shocked at 42 °C for 60 min in a shaking water bath. To recover the cells, we poured the transformation mixture into 1 L YPD medium pre-warmed to 30 °C and shook them at 30 °C for 4 hr. Recovered cells were spun at 5,000 g for 5 min, the liquid decanted, and the pellet resuspended in 8 mL PBS buffer. We plated the mixture on fourteen 150 × 15 mm plates (Fisher #08-757-14; 500 µL YPD + G418 per plate) and grew the cells at 30 °C for 2 d. A dilution series plated in parallel indicated 1.16 million cells per 500 µL plated cells, for a total of 16.2 million yeast transformants.

Plates were scraped with 5 mL PBS per plate, the harvested cells combined, spun (3,000 rpm for 5 min), and resuspended in 30 mL PBS. This cell suspension had an $OD_{600}$ of 128, as estimated by

measuring a dilution series. We inoculated 400 million cells (104 µL) in 250 mL YNB (MSG) + glucose + G418 medium, grew them at 30 °C for 26 hr, measured $OD_{600}$, and stored aliquots (750 µL culture in 750 µL 40% glycerol) at −80 °C. Each aliquot contained ~ 60 million cells.

The Upstream library was processed similarly, with the following differences. Transformation was performed using 320 µg of plasmid DNA and resulted in a total of 67 million transformants. About 100 OD units were harvested from the plates, and 1 billion cells (133 µL) grown overnight. Stored aliquots from this culture contained 250 million cells in 500 µL.

## Yeast growth

Samples were processed using two different sets of protocols. Briefly, the initial six TSS replicates were processed at smaller scale and with two-step PCR reactions. Below, this protocol is called 'Protocol 1'. The remaining TSS samples and all Upstream samples were processed at larger scale and using one-step PCR reactions ('Protocol 2'). See *Table 1—source data 1* for further details on each sample.

### Protocol 1

Aliquots of the glycerol stocks prepared during yeast transformation were thawed, spun down, the liquid removed, and the cells resuspended in 1 mL of YNB (MSG) + glucose + G418 medium. Resuspended cells were added to 100 mL medium per replicate and grown at 30 °C to an $OD_{600}$ of 0.4–0.5. At this OD, we harvested 50 million cells for DNA prep and 100 million cells for RNA prep, assuming 30 million cells per 1 mL culture at $OD_{600}$ = 1. Cells were spun down (5 min at 5,000 g), supernatant removed, and pellets placed on ice and stored at −80 °C as quickly as possible.

### Protocol 2

Aliquots of 500 million cells were inoculated in 650 mL of growth medium and grown to $OD_{600}$ = 0.4. From each replicate, we collected five pellets each from 40 to 50 mL of culture (~500 million cells each) for RNA extraction, and three to four pellets of 1–2 g of yeast for DNA extraction.

## DNA extraction
### Protocol 1

DNA was extracted using the Qiaprep Spin Miniprep kit (Qiagen, #27106) combined with an in-house yeast lysis protocol. To each sample of 50–60 million cells, we added 10 U zymolyase in 800 µL Y1 buffer (dissolve 182.2 g Sorbitol (Sigma Aldrich, #S6021) in 600 mL water, add 200 mL 0.5M $Na_2EDTA$, pH eight solution (Ambion, #AM9262), add 1 mL 14.3M β-mercaptoethanol (Sigma Aldrich, #M3148), adjust to 1 L with water) and lysed the cells at 30 °C for 30 min in a Thermomixer at 750 rpm. Cells stripped off their cell walls were spun for 10 min at 300 g, and the supernatant removed. The remaining steps followed the Qiagen protocol. DNA was quantified using Qubit dsDNA HS assays (ThermoFisher #Q32854).

### Protocol 2

DNA was extracted from 1 to 2 g of yeast using the Qiagen Plasmid Plus Midi kit, following a user-supplied Qiagen protocol ('QP11.doc Aug-01' available from Qiagen). We performed two 200 µL elutions. DNA was further purified and concentrated using bead clean-up (Kapa; #KK8001) and eluted in 50 µL TE buffer.

## RNA extraction
### Protocol 1

Total RNA was extracted from 100 million cell aliquots using the Zymo Research Fungal/Bacterial RNA Miniprep kit (ZR, #R2014) including DNase I treatment. Bead beating was performed on a mini bead beater (Biospec). mRNA was purified using the Dynabeads mRNA Purification kit (Ambion #61006) and eluted in 10 µL of water. Total RNA and mRNA were quantified using the Qubit RNA BR kit.

## Protocol 2

Total RNA was extracted using the Qiagen RNeasy Maxi Kit, with bead beating in 96-well format on the mini bead beater and on-column DNAse digestion. Total RNA was concentrated using Amicon Ultra-0.5 filter columns (Millipore #UFC503024).

# DNA sequencing library construction

## Protocol 1

We used PCR to amplify the barcodes from the extracted plasmids and to generate Illumina TruSeq-compatible sequencing libraries. The resulting libraries all carried the same i5 index (which we did not sequence) and sample-specific i7 indexes that allowed multiplexed sequencing. Library generation was performed as two PCRs (*Figure 1—figure supplement 6*). The first PCR used universal primers (without Illumina indexes) to amplify the barcodes. Each reaction contained 7.5–10 ng of plasmid DNA, 10 μL of Phusion polymerase buffer, 1 μL dNTPs (10 mM; Invitrogen #18427013), 0.25 μL primer 'common_ORF_v4' (100 μM), 0.25 μL primer 'RT_PCR_R_long' (100 μM), 0.5 μL Phusion HotStart II HF Polymerase, and water to 50 μL. Cycling conditions were 98 ℃ for 30 s, 15 cycles of 98 ℃ for 10 s, 55 ℃ for 30 s, 72 ℃ for 15 s, and a final extension at 72 ℃ for 5 min followed by a hold at 8 ℃. Reactions were cleaned up using Qiagen PCR purification kits, eluted in 30 μL EB, and quantified with Qubit HS DNA kits.

The 2nd PCR added sample-specific indexed primers. We adjusted the product of the first PCR to 25 ng in 10 μL, and assembled the same reaction mix as for the first PCR but with different primers: a sample-specific 'RT_PCR_D7xx_F' primer and the common 'Illumina_PCR_R' primer. The reaction was performed as above, but for six cycles and quantified with Qubit HS DNA kits.

## Protocol 2

We performed two-step PCR as above, with the following changes. Instead of Phusion, we used KAPA amplification mix (KK2602) for a reaction containing 25 μL 2X KAPA mix, 5 μL DNA, 19.5 μL water, 0.25 μL 'common_ORF_v2' (100 μM), and 0.25 μL primer 'RT_PCR_R_long' (100 μM). Cycling conditions, cleanup, quantification and adjustment to 25 ng / μL were the same as in Protocol 1. For the 2nd PCR, we performed two reactions per sample, which were then combined. For Upstream samples, we performed eight parallel reactions, in order to use the entire sample. PCR conditions were the same as for the second step PCR in Protocol 1, but with 1 μL of DNA and 23.5 μL water. Before pooling, individual samples were quantified using qPCR Library Quantification (KAPA, #KK4824).

For the 2018 Upstream samples (*Table 1—source data 1*), we performed single step PCR on the DNA samples using the same reaction mix as in the 1st step PCR above, but with primers 'RT_PCR_F_longD7xx' and 'RT-PCR-R-long'. PCR conditions were as above for the 1st PCR, but with 15 cycles.

# RNA-sequencing library construction

## Protocol 1

First-strand cDNA was synthesized by priming off the PS3 site immediately downstream of the barcodes. We combined 500 ng of mRNA in 4 μL water, 1 μL primer 'RT_PCR_R_long' (20 μM), 1 μL dNTPs (10 mM; Invitrogen #18427013), and 8.5 μL water, melted mRNA structure by incubating at 65 ℃ for 5 min, and placed the mixture on ice. We added 4 μL 5X reverse transcriptase buffer, 0.5 μL RiboLock RNAse inhibitor (ThermoFisher EO0381), and 1 μL Maxima Reverse Transcriptase (ThermoFisher EP0742), and incubated at 30 ℃ for 30 min and 85 ℃ for 5 min. To aid downstream PCR, we dissolved the cDNA/RNA hybrid by adding 1 μL ribonuclease H (Invitrogen 18021–014) and incubating at 37 ℃ for 20 min.

For second-strand cDNA synthesis, we performed a single extension using a forward primer that added a handle to be used in PCR. The corresponding handle for the first cDNA strand had already been added as part of the RT_PCR_R_long primer (*Figure 1—figure supplement 6*). We combined 5 μL of cDNA, 10 μL of 5X Phusion buffer, 1 μL dNTPs (10 mM; Invitrogen #18427013), 2.5 μL primer 'common_ORF_v4' (20 μM), 0.5 μL Phusion HotStart II HF Polymerase (ThermoFisher #F549L), and 31 μL water. We ran a single extension using the following thermocycler program: 98 ℃ for 30 s, 98℃ for 10 s, 55℃ for 30 s, 72℃ for 15 s, 72℃ for 5 min, and a hold at 8℃.

When present in a PCR reaction, the primers used above in cDNA synthesis would be capable of amplifying plasmid DNA. To prevent this, we degraded unincorporated primers by adding 5 µL of exonuclease I (ThermoFisher #EN0581), and incubating at 37 ˚C for 15 min and at 85 ˚C for 15 min. This last step at 85 ˚C inactivated the exonuclease but not the Phusion enzyme that was still present in the mixture, such that we were able to use the same mixture for additional PCR. To amplify the library and add i7 index primers, we added 0.25 µL primer 'Illumina_PCR_R' (100 µM) and 0.25 µL of a sample-specific 'RT_PCR_D7xx_F' primer (100 µM), and amplified as follows: 98 ˚C for 30 s, 18 cycles of 98 ˚C for 10 s, 55 ˚C for 30 s, 72 ˚C for 15 s, and a final extension at 72 ˚C for 5 min followed by a hold at 8 ˚C. The products were quantified with Qubit HS DNA kits.

### Protocol 2

First-strand cDNA synthesis was performed as in Protocol 1, with the following changes. We used SuperScript IV Reverse Transcriptase (Thermo Fisher, #18090–200) and incubated at 50 ˚C for 1 hr, using 2 µM 'RT_PCR_R_long' primer. To eliminate total RNA and cDNA/RNA hybrids after reverse transcription, we treated each cDNA sample with 1 µL RNase A (DNase and protease free, Thermo Fisher, #12091–021) and 1 µL RNase H (Thermo Fisher, #18021–071).

PCR was performed in a single step, using 5 µL 2x KAPA mix (Roche, #KK2602), 10 µL cDNA, 1 µL water, 0.25 µL sample-specific primer 'RT_PCR_F_long_D7xx' (100 µM), 0.25 µL common primer 'Illumina_PCR_R' (100 µM). PCR was performed as in Protocol 1, but for 15 cycles. For all Upstream as well as the 2018 TSS replicates, we performed parallel PCR reactions to capture as many cDNA molecules as possible. Before pooling, individual samples were quantified using qPCR Library Quantification (Roche, #KK4824).

## Pooling and sequencing

All PCR products from a given batch, in particular those made from DNA and cDNA from the same sample, were quantified (using Qubit or qPCR, see above) and pooled to equal molarity. Pooled libraries were gel extracted using QiaQuick Gel Extraction kits from 2% gels with 1% EtBr, run alongside a 50 bp DNA ladder. The libraries were visible in sharp bands narrowly centered on 198 bp. Excised pools were eluted in 30 µL EB buffer. Pooled libraries were quantified using qPCR Library Quantification (KAPA, #KK4824).

Sequencing was performed on Illumina HiSeq 2500 instruments in 'rapid mode' using 15–20% phiX spike-in, reading 50 bp single ends and the i7 index.

## Barcode counting

Sequences were demultiplexed using a custom php script that allowed up to one mismatch between the expected and the observed index. Indexes for each sample are available in *Table 1* – Source Data one and *Supplementary file 5*. In each sample, we counted the number of reads that contained a unique barcode. Only barcodes that had been previously observed in the annotation runs were retained. Barcode reads that did not perfectly match the annotation were excluded. While this also excluded reads with sequence errors that could in principle be added to the analyses, the resulting data loss was minor: across all DNA and RNA samples, a median of 85% (a range of 75–90%) of reads mapped to known barcodes. Barcode counts for technical replicates, in which multiple indexed libraries had been created from the same growth culture, were summed. As a metric for the expression driven by each promoter oligo, we considered the $\log_2$ ratio of summed RNA counts of all barcodes assigned to a given oligo, divided by the summed DNA counts. We used these ratios for visualization and comparison to other data, but not in significance testing (see below).

For two out of twelve TSS samples and two out of six Upstream samples, DNA counts were unavailable. We replaced these DNA counts with those from one other sample from the same batch (*Table 1* – Source Data 1). In the Upstream samples, we tested three additional strategies for addressing missing DNA values: replacement with the sum of all Upstream DNA samples, replacement with the counts from the Upstream annotation run, or sample exclusion. These different treatments did influence how many variants reached statistical significance, but only very slightly altered the estimated variant effect sizes (fold-changes). Eliminating the two Upstream samples with missing DNA would have resulted in fewer significant variants than we chose to report here, presumably due to reduced power (*Myint et al., 2019*). Among the options that retained all samples with RNA

counts, our choice of using DNA counts from another sample from the same batch resulted in the fewest significant variant effects. Thus, our treatment of missing DNA was conservative, while still allowing us to use RNA data from all available samples.

To compare the expression driven by promoter fragments to native gene expression in the genome, we computed the average expression driven by all fragments (irrespective of which allele they carried) designed from the promoter of the given gene.

Raw data and barcode counts are available under GEO accession GSE155944 (https://www.ncbi. nlm.nih.gov/geo/query/acc.cgi?acc=GSE155944).

## Statistical analyses of variant effects

Tests for differential allele activity were performed in the 'mpra' R package (*Myint et al., 2019*). At each variant, we performed pairwise comparisons of the expression driven by the RM allele versus that driven by the BY allele. When there were multiple variants within a promoter fragment assayed by the TSS library, we computed one test per variant such that the oligo carrying the RM allele for one particular variant was compared to the oligo carrying BY alleles at all variants.

We analyzed only those oligos that had summed barcode counts larger than zero in every replicate in both DNA and RNA data. After this filter, 2427 unique variants in 1429 promoters remained in the TSS library. In the Upstream library, 4467 variants in 1824 promoters remained.

We used the 'sum' barcode aggregation option in mpra, as well as mpra's normalization for library size. When a variant was assayed multiple times on a given strand in the two libraries, we used the smallest p-value and its associated fold change for downstream analyses. FDR was calculated by the mpra package (*Myint et al., 2019*).

To estimate the fraction of variants that have effects irrespective of their individual significance, we computed the $\pi_1$ statistic (*Storey and Tibshirani, 2003*). To do so, we used the qvalue package in R (*Storey et al., 2020*) to obtain the $\pi_0$ statistic (the proportion of statistical tests that are truly null) from the distribution of p-values of individual variants and then calculated $\pi_1 = 1 - \pi_0$.

## Comparison of variant effects and local eQTLs

Local eQTL effects were obtained from *Albert et al., 2018*. Effect sizes were expressed as log-fold changes of the RM allele relative to the BY allele. Because eQTLs can span wide genomic intervals, effect sizes and LOD scores of local eQTLs can differ slightly depending on whether they are determined at the peak marker of the given local eQTL or at the location of the gene itself. To use a metric that can be applied to all genes with MPRA data irrespective of whether the gene had a significant local eQTL, we used effects and LOD scores at the gene location. Confidence intervals for Spearman's rank correlation were computed using the DescTools package (*Signorell et al., 2020*).

Data on allele-specific mRNA expression in the BY/RM hybrid were obtained from Source Data seven in *Albert et al., 2018*, which comprised allele-specific expression data from two datasets (*Albert et al., 2018*; *Albert et al., 2014a*). We performed a Fisher's exact test on whether genes with at least one causal MPRA variant were more likely to show genome-wide significant allele-specific expression in at least one of the two datasets in *Albert et al., 2018*.

## Non-additive interactions

We tested for non-additive interactions among pairs of variants assayed in the TSS library. Specifically, we assayed 342 variant pairs in promoters for which exactly two variants were present in the promoter regions assayed by the TSS library. For these variant pairs, our design included four oligos that represented all possible combinations of the two alleles at the two given variants. We retained only promoters in which all four oligos had summed barcode counts larger than zero in DNA and RNA in every replicate.

To test for interactions, we used the mpra package (*Myint et al., 2019*) to fit the following model:

$$y = \beta_0 + \beta_1 x_1 + \beta_2 x_2 + \beta_3 x_1 x_2 + \epsilon$$

In the model, $y$ is the expression driven by an oligo, $x_1$ and $x_2$ are indicator variables relating an observation to the given genotype, $\beta_0$ is the intercept, $\beta_1$ and $\beta_2$ are the additive effects of the two

variants, $\beta_3$ is the interaction effect that we sought to test, and $\varepsilon$ is the residual error. We contrasted this model to a model without the $\beta_3 x_1 x_2$ term.

## Linkage disequilibrium

Pairwise linkage disequilibrium was calculated based on genotype data from a worldwide panel of yeast isolates (*Peter et al., 2018*). We computed the D' and $r^2$ linkage disequilibrium statistics as implemented in the 'genetics' R package (*Warnes, 2019*), using the two most frequent alleles at each marker.

## Variant annotation for non-TF features

We gathered 31 non-TF features to describe each variant (*Supplementary file 3*). Gene annotations were obtained from *Albert et al., 2018*. Nucleosome scores of a given variant in the genome were based on nucleosome occupancies reported in Supplementary Table 2 of *Brogaard et al., 2012*, by extending the positions of the reported nucleosome center by 72 nucleotides in both directions. Nucleosome-free regions were defined as those with no reported occupancy. The number of TATA box motifs were counted for each variant allele and its flanking sequence using the consensus TATA (A/T)A(A/T)(A/G) (*Basehoar et al., 2004*). Similarly, start codons were defined as occurrences of 'ATG'. Ancestral alleles were defined as those present in the Taiwanese soil strain EN14S01 ('standardized name': 'AMH') from the highly diverged clade 17, using the genotype data from *Peter et al., 2018*. We did not assign ancestral status for variants at which EN14S01 was heterozygous. Allele frequencies were obtained from genotypes in *Peter et al., 2018*. PhastCons scores, which quantify nucleotide conservation across seven *Saccharomyces* species (*Siepel et al., 2005*), were obtained for the positions corresponding to our variants from the UCSC genome browser (https://genome.ucsc.edu/index.html) (*Haeussler et al., 2019*).

## Quantifying variant effects on predicted transcription factor binding

We obtained position weight matrices (PWMs) for 196 TFs from the ScerTF database (*Spivak and Stormo, 2012*). For each variant and TF, we computed how well sequences containing the variant matched the TF's PWM. Higher 'TFBS scores' represent better matches between the sequence and the PWM, increasing the probability that the TF binds the given sequence.

For a given variant, we calculated TFBS scores in sliding windows of width equal to the length of the PWM, which we moved over the variant in one-base-pair increments. To compute the TFBS scores, we summed the position weights from the PWM for the bases dictated by the sequence of the given window. We computed these windowed TFBS scores for the BY and the RM allele. For each allele, we retained the best (i.e. the highest) and the mean TFBS score across windows. In the text, we report results based on the absolute difference between the best TFBS scores of the two alleles. Differences between the mean TFBS scores of the two alleles yielded similar results (*Figure 5—source data 1*).

For each TF, the ScerTF database provides sequence score cutoffs, above which the given sequence is considered to be a 'strong' match to the PWM. We deemed TFBS scores below these thresholds to be 'weak' TFBSs. We computed TFBS scores separately for strong and weak matches. For strong TFBSs, we considered only sequence windows in which the TFBS score exceeded the sequence score cutoff. We also computed the number of strong binding sites for each TF by counting the number of windows that exceeded the cutoff. For weak TFBSs, we only considered TFBS scores below the cutoffs for each TF.

Together, these analyses yielded five metrics for how a variant is predicted to perturb the binding of a given TF: the allelic difference in (1) the best and (2) the mean TFBS score across windows for strong TFBSs, (3) the number of strong binding sites, (4) the best and (5) the mean TFBS score across windows for weak TFBSs.

To consider strand-specificity in variant effects on TF-binding, these five allelic TFBS metrics were computed across three strand contexts, for a total of 15 features per TF: (1) the sense (or 'plus') strand, that is the 5' to 3' sequence of nucleotides upstream of the reporter gene; (2) the antisense (or 'minus') strand, by analyzing the reverse complement of the plus strand; (3) 'strand-agnostic', in which we computed the difference between the best and mean scores across windows irrespective

of which strand these scores came from. Across the 196 TFs, these features comprised a total set of 2940 features.

To obtain the aggregated TF features that summarized the 196 TF-specific sets of features, we computed the following 27 features for each variant. In each of the three strand contexts, we counted the total number of strong TFBSs changed (three features). Separately for strong and weak binding, we also computed the maximum difference in best or mean TFBS score across all TFs (12 features), and the average difference in best or mean TFBS score across all TFs (12 features).

To annotate individual TFBSs in *Figure 5—figure supplement 1*, we extracted the genome reference sequence flanking the two variants shown in the figure, up to a number of bases corresponding to the longest TF motif annotated in the Yeastract database (*Monteiro et al., 2020*), and then uploaded these sequences with either the BY or the RM allele to the Yeastract web server (http://www.yeastract.com).

## Logistic regression tests of single features

We used logistic regression to test if each feature influenced the classification of a variant as causal. Causal variants were defined as those variants detected at an FDR of 5% or better. We contrasted this set to a set of non-causal variants with an unadjusted p-value larger than 0.2. Variants in between these two categories were excluded from the analyses.

Features were transformed to Z-scores using the formula:

$$Z = (X_i - \mu i)/\sigma_i$$

Here, $X_i$ is the value of the feature for variant $i$, and $\mu_i$ and $\sigma_l$ refer to the mean and standard deviation of the feature values across all variants, respectively.

We performed logistic regression for each feature using the model:

$$S_i = \beta_0 + \beta_{library} x_{library} + \beta_{expression} x_{expression} + \beta_{feature} x_{feature} + \epsilon$$

Here, the response variable $S_i$ is a binary vector indicating whether variant $i$ is significant. We included the library a variant was measured in and the average normalized baseline expression driven by the oligos used to measure the variant as covariates in the models. $\beta_{library}$ and $\beta_{expression}$ are the effects of these two possible confounders, while $x_{library}$ and $x_{expression}$ are indicator variables relating an observation to the given covariate. $\beta_0$ is the overall intercept, and $\varepsilon$ is the residual error. $\beta_{feature}$ is the effect of the given feature on the probability that a variant is significant, and $x_{feature}$ is an indicator variable relating the feature values to the variants.

We used ANOVA to contrast this model to one without the $\beta_{feature} x_{feature}$ term. We corrected for multiple testing by computing FDR as q-values using the 'qvalue' package (*Storey and Tibshirani, 2003*).

## Multiple regression models

To build predictors of variant causality ('causal' vs 'non-causal'), we first divided our set of features into 112 partially redundant feature subsets. These 112 models differed in whether they considered the non-TF features, whether they included TF features from the plus, minus, and/or the strand-agnostic set, whether they considered strong and/or weak TF metrics, whether they included the aggregated TF summary features, and whether they included only features that had been significant in the single regression analyses. Details on each model are listed in *Figure 6—source data 1*.

For each of the 112 feature subsets, we built a logistic regression model to predict variant causality. For building these models, we first divided the variants also used in single-feature analyses into a training set comprising 90% of the data, and a test set comprising 10% of the data. Training was performed using repeated 10-fold cross validation as follows. The training set was split ten times, each time considering one of ten possible non-overlapping fractions of 10% of the training set as a validation set. In each of these ten splits, we used the remaining 90% of the training set to fit the model and computed Cohen's Kappa on the given validation set. This process was repeated five times, on five different 10-fold splits of the training data. The models were trained to maximize Cohen's Kappa on the validation sets. After training, each of the 112 models was applied to the 10% test set that had been held out from training. Model accuracy for the logistic regression models was measured by the area under the receiver operator characteristic curve (AUC). Elastic-net logistic

regression was performed using the 'caret' package (*Kuhn et al., 2020*). We excluded 35 features with an 'NA' value (27 aggregated TFBS features, expression level of the associated gene, two measures of dNdS, PPI, three measures of synthetic genetic interactions, nucleosome score).

Linear regression models to predict variant effects (expressed as absolute log-fold changes) were performed using the same repeated 10-fold cross-validation scheme and trained to minimize the root-mean squared error (RMSE). Prediction accuracy was measured using Spearman correlation coefficients between predicted and actual data on the test set. All model training was done using the 'caret' package (*Kuhn et al., 2020*).

In the case of multiple linear regressions, we also calculated model fit as the $R^2$ on the entire set of variants used in the single-feature regressions. These linear models were fit using the 'lm' function in R.

Elastic-net regression models predicting variant effects followed the same splitting and training strategy as for the 112 non-regularized models. We tuned the model across 10 by 10 combinations of the mixing parameter (alpha) and the regularization parameter (lambda) to select the model with the lowest RMSE in cross-validation. The best model was used to predict on the 10% test set, and model accuracy was measured as above. Elastic nets were trained using the 'caret' package (*Kuhn et al., 2020*).

## Predicting variant effects on our data using the *de Boer et al., 2020* model

To predict gene expression corresponding to our variants, we used the non-positional YPD model with pTpA embedding from *de Boer et al., 2020*. Among the models trained in that study, this model is least sensitive to the surrounding sequence context, which differs between our sequences and those used by de Boer et al. We used the model trained in YPD medium; the same medium we used in our experiments. The model uses 110 bp of promoter sequence as input to predict gene expression. Because our oligos were 144 bp long, we could not use the complete oligo sequence for prediction. Instead, we used 110 bp of DNA centered on each variant. For each variant, we used two 110 bp sequences, one containing the BY allele and the other containing the RM allele. The sequences were one-hot encoded using code from de Boer et al. and then fed into the model.

The predicted expression values for the BY and RM alleles of each variant were subtracted to predict fold change between alleles. Predicted expression and variant effects were compared to those observed in our experiments. As observed oligo expression values, we used the 'AveExpr' value in *Supplementary file 3* as the expression of the BY allele, and the sum of the 'AveExpr' and 'logFC' as the expression of the RM allele. The 'logFC' was used as the observed variant effect.

## Acknowledgements

We are grateful to Joshua Bloom, Chad Myers, and Henry Ward for input on data analysis. We thank Carl de Boer for help with applying his gene expression prediction model, Joshua Bloom for providing the BY/RM genotype data, and Liangke Gou for help with nucleosome data. We thank Suhua Feng for assistance with Illumina sequencing. We acknowledge resources and support from the Minnesota Supercomputing Institute.

## Additional information

### Funding

| Funder | Grant reference number | Author |
| --- | --- | --- |
| National Institutes of Health | R35GM124676 | Frank Wolfgang Albert |
| Howard Hughes Medical Institute | | Leonid Kruglyak |
| Pew Charitable Trusts | | Frank Wolfgang Albert |
| Alfred P. Sloan Foundation | | Frank Wolfgang Albert |
| Kinship Foundation | | Sriram Kosuri |

| Department of Energy, Labor and Economic Growth | DE-FC02-02ER63421 | Sriram Kosuri |
| National Institutes of Health | R01GM102308 | Leonid Kruglyak |
| National Institutes of Health | DP2GM114829 | Sriram Kosuri |

The funders had no role in study design, data collection and interpretation, or the decision to submit the work for publication.

## Author contributions

Kaushik Renganaath, Data curation, Software, Formal analysis, Investigation, Visualization, Methodology, Writing - original draft, Writing - review and editing; Rockie Chong, Conceptualization, Investigation, Methodology, Writing - review and editing; Laura Day, Data curation, Investigation, Methodology; Sriram Kosuri, Conceptualization, Resources, Supervision, Funding acquisition, Methodology, Writing - review and editing; Leonid Kruglyak, Conceptualization, Resources, Supervision, Funding acquisition, Writing - review and editing; Frank W Albert, Conceptualization, Data curation, Software, Formal analysis, Supervision, Funding acquisition, Validation, Investigation, Visualization, Methodology, Writing - original draft, Project administration, Writing - review and editing

## Author ORCIDs

Kaushik Renganaath https://orcid.org/0000-0003-1010-3604
Rockie Chong https://orcid.org/0000-0001-6736-9687
Sriram Kosuri http://orcid.org/0000-0002-4661-0600
Leonid Kruglyak https://orcid.org/0000-0002-8065-3057
Frank W Albert https://orcid.org/0000-0002-1380-8063

## Decision letter and Author response

Decision letter https://doi.org/10.7554/eLife.62669.sa1
Author response https://doi.org/10.7554/eLife.62669.sa2

# Additional files

## Supplementary files

- Supplementary file 1. Oligo design.
- Supplementary file 2. Oligo counts per replicate.
- Supplementary file 3. Non-TF features for each variant, along with statistical results for each variant.
- Supplementary file 4. All features used in variant annotation, including TFBS. (gzipped text file).
- Supplementary file 5. Primers sequences and sequences of various components of the reporter gene construct.
- Supplementary file 6. BY/RM sequence variants used in MPRA design (gzipped vcf file).
- Supplementary file 7. Sequence and map of a plasmid in the library after completed library construction. In place of the library, the sequence contains an example promoter fragment.
- Transparent reporting form

## Data availability

Raw data and barcode assignments to oligos are available under GEO accession GSE155944. Source Data is provided for Figures 2, 3, 4, 5, and 6. Additional processed data and the MPRA design are available as Supplementary Files.

The following dataset was generated:

| Author(s) | Year | Dataset title | Dataset URL | Database and Identifier |
|---|---|---|---|---|
| Renganaath, Chong | 2020 | Massively parallel identification of cis-regulatory variants in yeast promoters | https://www.ncbi.nlm.nih.gov/geo/query/acc.cgi?acc=GSE155944 | NCBI Gene Expression Omnibus, GSE155944 |

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
