## [Decision Letter]

**Acceptance summary:**

The authors devised a new approach based on oligo synthesis, a transcriptional reporter and barcode sequencing to identify likely causal variants underlying cis regulatory variation in yeast. The results show that some promoter regions often have multiple SNPs affecting gene expression. The authors find that some of these regulatory SNPs show epistatic interactions and that natural selection may keep regulatory SNPs at low frequency in natural populations. SNPs affecting gene expression are enriched in known transcription factor binding sites. This study is a spectacular example of how the combination of emerging and established technologies can be exploited to gain a refined picture of genotype-phenotype maps and this, genome wide.

**Decision letter after peer review:**

Thank you for submitting your article "Massively parallel identification of causal variants underlying gene expression differences in a yeast cross" for consideration by *eLife*. Your article has been reviewed by three peer reviewers, and the evaluation has been overseen by a Reviewing Editor and Patricia Wittkopp as the Senior Editor. The reviewers have opted to remain anonymous.

The reviewers have discussed the reviews with one another and the Reviewing Editor has drafted this decision to help you prepare a revised submission.

Summary:

The authors devised a new approach based on oligo synthesis, a transcriptional reporter and barcode sequencing to identify causal variants underlying cis regulatory variation in yeast. The results show that some promoter regions often have multiple SNPs affecting gene expression. The authors find that some of these regulatory SNPs show epistatic interactions and that natural selection may keep regulatory SNPs at low frequency in natural populations. SNPs affecting gene expression are also enriched in known transcription factor binding sites.

The three reviews are highly positive. However, for the paper to be considered further, it would be important to perform some additional analyses and revise some of the interpretations of the results. This is particularly the case for some of the recommended analyses aimed at disentangling the relationships between different factors associating with regulatory SNPs because these often covary. If it is impossible to completely estimate their independent contribution, this issue should at least be addressed in the Results and Discussion.

I have kept the three full reports below because they are complementary and well detailed. We expect no additional experiments at this point.

*Reviewer #1:*

This study investigates the molecular origin of differential transcription in the well-studied BY-RM cross. SNPs and indels in promoters between these two strains were reciprocally exchanged in order to measure their effects by sequencing barcoded transcripts. The authors use this method to do a deep dive into the genetic determinants of local eQTLs, in which they identify causal variants and give examples of proximal variants with non-additive interactions. They also explore the nature of the causal variants and work to predict variants and expression. In my opinion, this article is very well written, well measured, well explained, and well thought through. I have no major comments.

*Reviewer #2:*

The paper by Renganaath et al. uses a reporter assay and Illumina sequencing to estimate the effects on gene expression of thousands of naturally occurring promoter variants between two yeast strains. They identify a large number of variants that have significant effects on expression, greatly expanding the catalog of known individual regulatory variants. They use this catalog to test long standing ideas about the molecular and evolutionary nature of these regulatory variants. Overall, the experiments use a good design with the necessary controls and replication to identify variants with moderate and large effects on gene expression. In general I think the work makes a good contribution to the field and I only have a few comments and concern about the models used and the strength of the conclusions made from these models.

1) First, the authors have a number of potential explanatory variables and test each one individually for association with whether a variant significantly alters gene expression or not. The results from these regression analyses are taken as is, with no attempt to account for correlations among the factors themselves. The authors seem to be aware of this issue; one of the largest individual correlates is gene essentiality which the authors note is often associated with some of the other covariates used. Because of this issue, significant associations cannot be interpreted as meaning a particular covariate is important, and causal connections can't be made from this kind of analysis. Furthermore, the authors take a number of significant covariates and interpret them as being consistent with negative selection, but whether these covariates are all significant once others are accounted for is unknown. The different covariates may be detecting similar signals, in which case they are not independent. A more comprehensive modeling scheme would allow the most important covariates to be identified and lead to a better understanding of what signals actually exist in the data. For example, using regularization techniques (which the authors do use later in another analysis) on a model including all of the covariates would help to avoid non-independent covariates.

2) Similar arguments can be made for the regression analyses that incorporate transcription factor binding; the PWMs of TFs are not independent and many have similarities due to similar modes of binding. In addition, the data does not clearly show 'evidence that causal variations often perturb TF binding'. Instead, the data shows that variants predicted to alter the binding of TFs are correlated with whether a variant is causal. Again, direct causality from this type of analysis is difficult to do. This is made even more difficult to follow with the “weak” TF binding as it appears that the authors are arguing for a model where individual single nucleotide variants affect expression by altering the binding of multiple TFs, each of which has a low individual probability of being bound. While it is well known that many weak TFBS are present in DNA, I'm not familiar with changes to these weak TFBS being proposed in the literature as a major route by which gene expression is altered in nature. Two things could help make this claim stronger. First, it would help to see a clear example that was found in the data, e.g. of a causal variant that was predicted to alter a single strong TFBS and one that was predicted to affect multiple weak TFBS. Second, functional validation of some of these variants as affecting the predicted TF binding (and not some other, unknown and untested factor) would significantly increase the impact of this section. This later route would likely require substantial effort, but at the very least, that such functional analysis has not been done needs to be recognized and the claims moderated in this section.

*Reviewer #3:*

Renganaath and colleagues described a high-throughput assay for testing how natural genetic variants influence gene expression, as well as finding patterns that allow us to predict such effects. The main novelty of their approach is in its single-variant resolution. Other methods such as eQTL mapping and allele-specific expression (ASE) do not have this level of resolution, so MPRA is a valuable addition to the yeast genetics toolkit. The ability to test epistatic interactions of neighboring variants is also a nice feature.

1) The role of individual variants in contributing to ASE is key to understanding cis-regulatory logic. The authors designed MPRA oligos focusing on previously identified ASE genes and added randomly selected genes as controls. It would be interesting to see an aggregated statistic on whether ASE genes are more likely to harbor causal variants than non-ASE genes; and if upstream variants differ from TSS variants in such patterns.

2) Prediction of causal regulatory variants is a holy grail of functional genomics. In this study, the predictions show slightly (but significantly) better than random performance. It is worth noting that there are more non-causal variants that causal variants, resulting in imbalanced classification problem. Oversampling of the minority labels could be a good start to balance data distribution and improve prediction performance.

---

## [Author Response]

Reviewer #2:The paper by Renganaath et al. uses a reporter assay and Illumina sequencing to estimate the effects on gene expression of thousands of naturally occurring promoter variants between two yeast strains. They identify a large number of variants that have significant effects on expression, greatly expanding the catalog of known individual regulatory variants. They use this catalog to test long standing ideas about the molecular and evolutionary nature of these regulatory variants. Overall, the experiments use a good design with the necessary controls and replication to identify variants with moderate and large effects on gene expression. In general I think the work makes a good contribution to the field and I only have a few comments and concern about the models used and the strength of the conclusions made from these models.1) First, the authors have a number of potential explanatory variables and test each one individually for association with whether a variant significantly alters gene expression or not. The results from these regression analyses are taken as is, with no attempt to account for correlations among the factors themselves. The authors seem to be aware of this issue; one of the largest individual correlates is gene essentiality which the authors note is often associated with some of the other covariates used. Because of this issue, significant associations cannot be interpreted as meaning a particular covariate is important, and causal connections can't be made from this kind of analysis. Furthermore, the authors take a number of significant covariates and interpret them as being consistent with negative selection, but whether these covariates are all significant once others are accounted for is unknown. The different covariates may be detecting similar signals, in which case they are not independent. A more comprehensive modeling scheme would allow the most important covariates to be identified and lead to a better understanding of what signals actually exist in the data. For example, using regularization techniques (which the authors do use later in another analysis) on a model including all of the covariates would help to avoid non-independent covariates.

We agree with the reviewer that many of our features are correlated and want to stress that in the single-feature analyses the reviewer refers to, we did not intend to identify individual features that contribute to variant causality independently of all other features. Instead, in our view, when multiple related features are associated with causality, this hints at the influence of an underlying phenomenon that may not be perfectly captured by any one feature alone. To stay with the example highlighted by the reviewer, the associations of variant causality with 1) lower derived allele frequency, 2) occurrence in the promoters of non-essential genes, and 3) occurrence in the promoters of genes with fewer synthetic genetic interactions, are all consistent with an underlying signal of negative selection against variants that alter the expression of essential genes.

To make this point obvious to the reader, we have added the following statement: “While these analyses cannot isolate the individual contributions of features that are correlated with each other, they provide an overview of the characteristics of causal variants.”

Following the reviewer’s suggestion of attempting to account for correlated features via regularization, we have extended our existing linear elastic net models (in which we had modeled variant effect size as a quantitative response variable) to logistic elastic net models of whether or not a variant is causal. Briefly, we fit all features in one model with 10-fold repeated cross-validation (5 repeats). This model predicted variant causality fairly poorly, with an area under the ROC of 0.53 on a 10% held-out test set. In this model, 1,812 predictors had a non-zero importance score. We also fit our 112 logistic regression models of various feature subsets with elastic net regularization. The best of these models, which included non-TF features and individual TF features in the strand agnostic configuration that had been significant in the univariate analyses, achieved an AUC-ROC of 0.71, identical to the best model without regularization. We have added these results to the paper (subsection “Prediction of causal variants”).

We draw two conclusions from these results. First, regularization alone does not improve predictions. We suspect that our dataset of a few hundred causal variants, each in a different promoter context, is not large enough to permit more accurate predictions. Second, and more directly to the reviewer’s concern, the feature associations we detected in our univariate logistic regressions do not trivially collapse to a small set of highly correlated features.

2) Similar arguments can be made for the regression analyses that incorporate transcription factor binding; the PWMs of TFs are not independent and many have similarities due to similar modes of binding. In addition, the data does not clearly show “evidence that causal variations often perturb TF binding”. Instead, the data shows that variants predicted to alter the binding of TFs are correlated with whether a variant is causal. Again, direct causality from this type of analysis is difficult to do.

We agree that our concluding statement for this section (“…provided clear evidence that causal variations often perturb TF binding”) was worded too strongly. Closely following the reviewer’s suggestion, we now conclude this section with “Overall, these analyses provided clear evidence that variants predicted to perturb TF binding are more likely to alter gene expression than variants not predicted to do so.”

As we pointed out in response to the reviewer’s first comment above, our intention in these analyses was not to assign independent contributions to binding of individual factors. As the reviewer points out, doing so would be extremely challenging due to sharing of motifs by different factors. To make this point clear for readers, we have added the statement “These analyses cannot disambiguate sharing of similar binding motifs by different TFs but probe the overall role of perturbed TF binding in variant causality.” Together with our reworded concluding sentence, we believe that this clarifies how we interpret the results on altered TFBSs.

This is made even more difficult to follow with the “weak” TF binding as it appears that the authors are arguing for a model where individual single nucleotide variants affect expression by altering the binding of multiple TFs, each of which has a low individual probability of being bound. While it is well known that many weak TFBS are present in DNA, I'm not familiar with changes to these weak TFBS being proposed in the literature as a major route by which gene expression is altered in nature. Two things could help make this claim stronger. First, it would help to see a clear example that was found in the data, e.g. of a causal variant that was predicted to alter a single strong TFBS and one that was predicted to affect multiple weak TFBS. Second, functional validation of some of these variants as affecting the predicted TF binding (and not some other, unknown and untested factor) would significantly increase the impact of this section. This later route would likely require substantial effort, but at the very least, that such functional analysis has not been done needs to be recognized and the claims moderated in this section.

The reviewer is correct that we argue for “a model where individual single nucleotide variants affect expression by altering the binding of multiple TFs, each of which has a low individual probability of being bound.”, although we do not strongly argue that such variants always have to alter binding of *multiple* TFs. Note that in the Discussion, we wrote “[variants] may perturb one or multiple weak binding sites”.

This finding of strong association of causal variants with changes to weak TFBSs is supported by a recent study by de Boer et al., 2020, which we repeatedly cite in our Results and Discussion sections. The authors conducted a massively parallel reporter assay of more than 100 million random promoter fragments to build a highly predictive model of transcriptional regulation of RNA expression. In in-silico experiments probing their model, the authors set the level of individual TFs to zero and tested the influence of these “deletions” on gene expression. Strong regulatory effects (i.e., those that altered expression by more than 2-fold) of individual transcription factors were rare and accounted for only ~6% of the average expression driven by a typical promoter fragment. By contrast, the remaining 94% of the expression driven by a typical promoter fragment were attributed to the vast majority (99.9%) of interactions between a TF and a fragment that were individually weak. A key conclusion of the de Boer et al. paper is that (emphasis added) “[…] random DNA has diverse expression levels (Figure 1) that can be explained by TF binding (Figure 2), which regulate expression primarily through weak interactions (Figure  6) that, in turn, can easily be perturbed [by changes to DNA sequence] (Figure 5).”. Here, we extend this reasoning by de Boer et al. to natural variants.

To help the reader understand these points about weak binding sites, we have added a new Figure 5—figure supplement 1, which shows an example of a variant with a clear change to a strong TFBS (panel A), as well as a variant that does not change any recognizable strong TFBSs, but does change predicted binding at multiple weak TFBSs (panel B).

We do acknowledge that additional work (which is outside of the scope of the current manuscript) will be required for a fuller understanding of the role of weak binding sites in regulatory variation. We have modified the Discussion section to make this point more clearly.

Reviewer #3:Renganaath and colleagues described a high-throughput assay for testing how natural genetic variants influence gene expression, as well as finding patterns that allow us to predict such effects. The main novelty of their approach is in its single-variant resolution. Other methods such as eQTL mapping and allele-specific expression (ASE) do not have this level of resolution, so MPRA is a valuable addition to the yeast genetics toolkit. The ability to test epistatic interactions of neighboring variants is also a nice feature.1) The role of individual variants in contributing to ASE is key to understanding cis-regulatory logic. The authors designed MPRA oligos focusing on previously identified ASE genes and added randomly selected genes as controls. It would be interesting to see an aggregated statistic on whether ASE genes are more likely to harbor causal variants than non-ASE genes; and if upstream variants differ from TSS variants in such patterns.

Thanks to the reviewer for this suggestion. We turned to ASE mRNA data previously gathered and integrated with local eQTL results in Albert et al., 2018. Those data contain two independent BY/RM hybrid ASE datasets. Here, genes that had shown genome-wide significant ASE in at least one of these two datasets were more likely to have a causal MPRA variant (Fisher’s exact test: odds ratio = 3.2, p < 2.2e-16). We have added this result to the Results section and a corresponding Materials and methods paragraph.

This agreement between ASE data and our MPRA assay is further bolstered by a correlation between the number of causal MPRA variants and either the number of ASE datasets with a significant result for the given gene (rho = 0.23, p < 2.2e-16) or the absolute magnitude of ASE (rho = 0.14, p = 2e-8). These results were similar for variants in the TSS and Upstream MPRA libraries (Author response table 1).

**Author response table 1. resptable1:** 

Library	Fisher’s exact test (odds ratio)	Fisher’s exact test (p-value)	Correlation w/ number significant ASE datasets (rho)	Correlation w/ number significant ASE datasets (p-value)	Correlation with ASE magnitude (rho)	Correlation with ASE magnitude (p-value)
TSS	3.1	3e-7	0.18	2e-8	0.11	0.0009
Upstream	2.8	6e-10	0.21	3e-12	0.13	0.00001

In the interest of brevity, we chose not to include these more detailed analyses in the paper. Note that the ASE data we used in the analyses above differ slightly from those we had used for designing the MPRA. This is simply because more ASE data has become available since we designed the MPRA.

2) Prediction of causal regulatory variants is a holy grail of functional genomics. In this study, the predictions show slightly (but significantly) better than random performance. It is worth noting that there are more non-causal variants that causal variants, resulting in imbalanced classification problem. Oversampling of the minority labels could be a good start to balance data distribution and improve prediction performance.

We acknowledge the class imbalance in the training sets of our classifiers and thank the reviewer for this suggestion. To test if oversampling improves predictions, we oversampled causal variants to an equal representation of causal and non-causal variants in the training set used to train our classifiers. As expected, the classifiers with oversampling of the minority class were more stable during training and performed better in predicting on the *validation* sets: across the 112 models, median average Cohen’s Kappa increased from 0.02 without oversampling to 0.22 with oversampling (see the top left panel in Author response image 1; the red line denotes equality). There was a marginal improvement in the AUC-ROC when predicting on the *test* set (the best model improved from 0.71 to 0.73; see the top right panel in Author response image 1; this best classifier used the same features in the training stage as in our analyses in the paper). However, the average performance across the 112 models remained similar (bottom panel; the red line corresponds to equal performance; note that the points are symmetric around this line indicating that there is no overall trend for better performance). Given this absence of a systematic improvement, we have chosen not to include these results in the paper.

**Author response image 1. sa2fig1:**